# LAST-ITERATE CONVERGENCE OF ADMM ON MULTI-AFFINE QUADRATIC EQUALITY CONSTRAINED PROBLEM

## ABSTRACT

In this paper, we study a class of non-convex optimization problems known as multi-affine quadratic equality constrained problems, which appear in various applications–from generating feasible force trajectories in robotic locomotion and manipulation to training neural networks. Although these problems are generally non-convex, they exhibit convexity or related properties when all variables except one are fixed. Under mild assumptions, we prove that the alternating direction method of multipliers (ADMM) converges when applied to this class of problems. Furthermore, when the "degree" of non-convexity in the constraints remains within certain bounds, we show that ADMM achieves a linear convergence rate. We validate our theoretical results through practical examples in robotic locomotion.

## 1 INTRODUCTION

Non-convex optimization serves as a fundamental concept in modern machine learning, such as reinforcement learning Xu et al. (2021); Wang et al. (2024) and large language models Ling et al. (2024); Kou et al. (2024). The non-convexity in these applications may arise from the objective function, the constraint set, or both. Finding a solution to a non-convex problem is, in general, NP-hard Krentel (1986). As a step to manage this complexity, a common practice is to study problems with additional structural assumptions under which particular solvers, such as gradient-based methods, are guaranteed to converge to an optimizer. Subsequently, various relaxations of the objective and/or constraints have been proposed to transform the original problem into a more tractable problem. For instance, the objective function has been studied under assumptions such as weak strong convexity Liu et al. (2014), restricted secant inequality Zhang & Yin (2013), error bound Cannelli et al. (2020), and quadratic growth Rebjock & Boumal (2024). On the other hand, optimization problems with various types of non-linear constraints have been investigated, such as quadratically constrained quadratic programs (QCQP) Bao et al. (2011); Elloumi & Lambert (2019), geometric programming (GP) Boyd et al. (2007); Xu (2014), mixed-integer nonlinear programming (MINLP) Lee & Leyffer (2011); Sahinidis (2019), and equilibrium constraints problem Yuan & Ghanem (2016); Su (2023).

Recently, there has been growing interest in analyzing non-convex optimization problems with specific block structures, driven by their broad range of applications. Although such problems are generally non-convex, they often exhibit convexity or related properties when all but one block of variables is fixed. Various structural properties of these problems have been studied, including multi-convexity in minimization settings Xu & Yin (2013); Shen et al. (2017); Lyu (2024), PL-strongly concave Guo et al. (2023), and PL-PL Daskalakis et al. (2020); Chen et al. (2022) in min-max formulations. Motivated by two well-known applications in robotics, in this work, we study multi-affine equality-constrained optimization problems (see Problem in equation 1).

In particular, locomotion and manipulation problems in robotics (Figure 1) involve intermittent contact interactions with the world. Due to the hybrid nature of these interactions, generating dynamically-consistent trajectories for such systems leads to a set of non-convex problems, which remains an open challenge. In general, the problem of planning through contact is handled in two ways; contact-implicit and contact-explicit. The first approach directly incorporates the complementarity constraints arising from the contacts, either by relaxing them within the problem formulation Tassa et al. (2012) or at the solver level Posa et al. (2014). While this approach has recently shown considerable promise

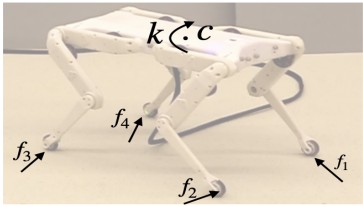 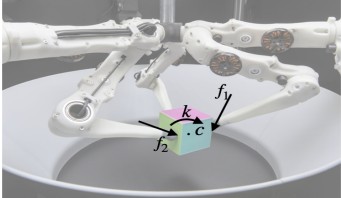

Figure 1: Examples of locomotion and manipulation settings, (left) Solo Grimminger et al. (2020) and (right) Trifinger Wüthrich et al. (2020)

in practice Kim et al. (2023); Aydinoglu et al. (2024); Le Cleac'h et al. (2024), providing convergence guarantees remains an open problem due to the presence of multiple sources of non-convexity. The second approach handles contact in the trajectory optimization problem by casting it as a mixed-integer optimization Deits & Tedrake (2014); Toussaint et al. (2018). In this approach, the hybrid nature of interaction is explicitly taken into account with integer variables, and thus, a combinatorial search is required to decide over the integer decision variables, while the continuous trajectory optimization problem ensures the kinematic and dynamic feasibility of the problem. This approach has also shown great success in recent years in both locomotion Ponton et al. (2021); Taouil et al. (2024); Aceituno-Cabezas et al. (2017) and contact-rich manipulation tasks Hogan & Rodriguez (2020); Toussaint et al. (2022); Zhu et al. (2023).

The optimization problem in the contact-explicit setting exhibits additional interesting structure. In particular, the dynamics can be decomposed into underactuated and actuated components Wieber (2006). This implies that, to generate a feasible force trajectory, the kinematics can be abstracted away. Assuming the robot can produce any desired contact force, it is sufficient to consider only the Newton-Euler dynamics to generate dynamically consistent trajectories for the robot's center of mass (CoM) in locomotion and for the object's CoM in manipulation. Interestingly, in this setting, the non-convexity in the dynamics has a special form, namely it is multi-affine Herzog et al. (2016). This renders the trajectory optimization problem a multi-affine equality-constrained optimization problem. These types of problems also appear in other applications such as matrix factorization Luo et al. (2020); Choquette-Choo et al. (2023), graph theory, and neural network training process Taylor et al. (2016); Zeng et al. (2021).

Recent work has exploited this structure to solve the problem using methods such as block coordinate descent Shah et al. (2021) and the alternating direction method of multipliers (ADMM) Meduri et al. (2023). In constrained optimization problems with linearly separable constraints, ADMM is an efficient and reliable algorithm Lin et al. (2015a); Deng et al. (2017); Yashtini (2021). Notably, its variants are driving the success of many machine learning applications involving optimization problems with linear constraints and convex objectives Shi et al. (2014); Nishihara et al. (2015); Khatana & Salapaka (2022) as well as those with non-convex objectives Boţ & Nguyen (2020); Kong & Monteiro (2024); Wang et al. (2019); Li et al. (2024); Yuan (2025). However, not all of these works provide convergence guarantees, and those that do either rely on additional assumptions or establish weaker forms of convergence. For instance, Boţ & Nguyen (2020) considered the problem $\min f(x) + \phi(z)$ subject to $Ax = z$, where $\phi$ is proper and lower semicontinuous (LSC), $h$ is differentiable and $L$-smooth. They proved last-iterate convergence to a KKT point under the KL property (see Definition 2.5) and assuming access to a proximal solver, i.e., $\arg\min_z\{\phi(z) + \langle y^k, Ax^k - z \rangle + \frac{r}{2}\|Ax^k - z\|^2 + \frac{1}{2}\|z - z^k\|^2\}$. A similar assumption is made by Yuan (2025). However, this requirement is quite restrictive when $g$ is nonconvex and the constraints are nonlinear. In fact, their guarantees no longer hold when the constraint set includes nonlinear relations, such as $\{x_1 x_2 + x_3 + z = 0\}$. The next table presents a comparison of existing ADMM approaches.

Naturally, it is important to understand the limitations of ADMM in settings with nonlinear constraints, such as multi-affine equality-constrained Wang et al. (2019); Zhang et al. (2023); Barber & Sidky (2024). El Bourkhissi & Necoara (2025) studied general nonlinear equality-constrained nonconvex optimization but imposed additional structural assumptions to ensure regularity, such as full column rank of the constraint Jacobian. They also relied on backtracking strategies to solve each subproblem. Furthermore, they established convergence rates under the assumption that the objective satisfies the $\alpha$-PL property. Li & Yuan (2025) focused on proximal ADMM methods for general nonlinear

Table 1: Comparison with previous work

| Ref. | Blocks | Objective | Assumptions | Constraints | Convergence |
|------|--------|-----------|-------------|-------------|-------------|
| Boţ & Nguyen (2020) | =2 | $f(x) + \phi(z)$ | $f$: smooth, $\phi$: LSC | Linear | Last iterate assuming $\alpha$-PL |
| Wang et al. (2019) | $\geq 2$ | $f(x) + \sum_{i=1}^{n} I_i(x_i) + \phi(z)$ | $f$: smooth, $\phi$: smooth $f + \phi$ : coercive | Linear | Avg. iterate |
| Li et al. (2024) | =2 | $f(x) + \phi(z)$ | $f$: smooth, $\phi$: convex, nonsmooth | Convex, bounded | Best iterate |
| Yuan (2025) | $\geq 2$ | $\sum_{i=1}^{n} f_i(x_i) + h_i(x_i)$ | $f_i$: smooth, bounded derivative, $h_i$ : nonsmooth, $h_n$ : convex | Linear | Avg. iterate |
| Our Work | $\geq 2$ | $f(x) + \sum_{i=1}^{n} I_i(x_i) + \phi(z)$ | $f$: strongly convex, smooth, $\phi$: strongly convex, smooth | Multi-affine, unbounded | Last iterate |

dynamics constraints. However, they only provide the best-iterate convergence between two iterates and fail to provide convergence rate related to the stationary point. In problems with multi-affine constraints, Gao et al. (2020) investigated the convergence properties of ADMM under the strict set of assumptions for the objective function (it has to be independent of certain variables). However, they fail to show any explicit convergence rate, which is crucial for applications like trajectory planning in robotics, where solutions must be computed within a short time slot with predefined accuracy. This leads to a key question that we seek to address in this study.

> *What convergence rate can be guaranteed for ADMM when applied to optimization problems with multi-affine quadratic equality constraints?*

Previous works have shown that a linear convergence rate is achievable in linearly constrained problems with a strongly convex objective Lin et al. (2015b); Cai et al. (2017); Lin et al. (2018). On the other hand, it is straightforward to see that the multi-affine quadratic constraints gradually reduce to linear constraints as the non-convex coefficients vanish (i.e., $\{C_i\} \to \mathbf{0}$ in equation 2). Thus, in the extreme case when all the non-convex coefficients are zero, linear convergence is ensured. However, our empirical results indicate that a linear convergence rate remains attainable even when the nonlinear quadratic components are present. This observation motivates our next research question.

> *If the effect of non-convexity in the constraint is small enough, does the linear convergence of ADMM when applied to the problem in 1 still hold?*

In this paper, we provide positive answers to both of the questions raised above. More precisely, when the norm of the non-convex coefficients (i.e., $\{\|C_i\|\}$) is sufficiently small relative to the norms of the linear components in the constraint set, ADMM achieves a linear convergence rate. Otherwise, under certain mild assumptions, we show that the convergence rate is sub-linear. In addition, we validate our theoretical findings through several practical experiments in robotic applications.

## 2 PROBLEM SETTING

**Notations:** Throughout this work, we denote $\|B\|$ and $\|a\|$ as the spectral norm of matrix $B$ and the Euclidean norm of vector $a$, respectively and denote the smallest eigenvalue of $B$ by $\lambda_{min}(B)$. We use $x_i$ and $x_{-i}$ to denote the $i$-th entry of the vector $x$ and all entries except the $i$-th entry, respectively. We denote $[x_i, x_{i+1}, \cdots, x_j]$ by $x_{i:j}$ when $j \geq i$ and the empty set when $j < i$. The partial derivative of function $f(x)$ with respect to the variables in its $i$-th block is denoted as $\nabla_i f(x) := \frac{\partial}{\partial x_i} f(x_i, x_{-i})$ and the full gradient is denoted as $\nabla f(x)$. The general sub-gradient of $f$ at $x$ is denoted by $\partial f(x)$. The exterior product between vectors $a$ and $b$ is $a \times b$. The set of all functions that are $n$-th order differentiable is $C^n$. The ball centered at $x$ with radius $r$ is $\mathcal{B}(x; r)$. The distance between a point $x$ and a closed set $\mathbb{S}$ is given by $dist(x, \mathbb{S}) := \inf_{s \in \mathbb{S}} \|s - x\|$. $\lambda_{min}(A)$ and $\lambda_{min}^+(A)$ denote the minimum eigenvalue and the minimum positive eigenvalue of $A$, respectively.

**Definitions and assumptions:** In this work, we consider the following *multi-affine quadratic equality constrained* problem:

$$\min_{x,z} F(x) + \phi(z), \quad \text{s.t.} \quad A(x) + Qz = 0, \tag{1}$$

where $x = (x_1, \cdots, x_n)^T \in \mathbb{R}^{n_x}$ is partitioned into $n$ blocks, with each block $x_i \in \mathbb{R}^{n_i}$. $Q$ is a matrix in $\mathbb{R}^{n_c \times n_z}$, and $z$ is a vector in $\mathbb{R}^{n_z}$. Function $A(x)$ is a multi-affine quadratic operator.

> **Definition 2.1.** *Function $A(\cdot) : \mathbb{R}^{n_x} \to \mathbb{R}^{n_c}$ is called a multi-affine quadratic operator when for each $i \in \{1, ..., n_c\}$, there exist $C_i \in \mathbb{R}^{n_x \times n_x}$, $d_i \in \mathbb{R}^{n_x}$, and $e_i \in \mathbb{R}$ such that*
>
> $$(A(x))_i := \frac{x^T C_i x}{2} + d_i^T x + e_i, \tag{2}$$
>
> *and moreover, $A(x_j; x_{-j})$ is an affine function for $x_j$ when $x_{-j}$ are fixed, $\forall x_{-j}, \ j \in [n]$.*

Note that the set of constraints in equation 1 comprises the linear ones and encompasses a much broader class in nonlinear settings. It also appears in various applications such as the locomotion and manipulation problems in robotics, matrix factorization, and neural network training process. From definition, it is obvious that the diagonal blocks of the matrix $C_i$ is zero matrix. We provide a simple example of the multi-affine quadratic equality constrained problem.

**Example 2.2.** *Consider the following problem*

$$\min_{x,z} x_1^2 + x_2^2 + z_1^2 + z_2^2, \quad s.t. \quad x_1 x_2 + x_1 + 1 + z_1 = 0, \quad -x_1 x_2 + x_2 + 1 + z_2 = 0.$$

*This problem can be reformulated in the form of equation 1 by considering $Q$ to be the identity matrix, $F(x) := x_1^2 + x_2^2$, $\phi(z) := z_1^2 + z_2^2$, and*

$$C_1 = \begin{bmatrix} 0 & 1 \\ 1 & 0 \end{bmatrix}, \quad C_2 = \begin{bmatrix} 0 & -1 \\ -1 & 0 \end{bmatrix}, \quad d_1 = \begin{bmatrix} 1 \\ 0 \end{bmatrix}, \quad d_2 = \begin{bmatrix} 0 \\ 1 \end{bmatrix}, \quad e_1 = 1, \quad e_2 = 1.$$

The next set of assumptions are made to restrict the objective function in equation 1. Namely, we assume that the function $F(x)$ can be decomposed into a $C^2$ strongly convex function and a group of indicator functions that are block-separable.

**Assumption 2.3.** *$F(x)$ is subanalytic (see Appendix A for formal definition) and can be written as $f(x) + \sum_{i=1}^n I_i(x_i)$, where $f(x)$ is $C^2$, $\mu_f$-strongly convex with $x_f^\star$ denoting its minimizer and $I_i(\cdot)$ is the indicator function of a convex and closed set $X_i \subseteq \mathbb{R}^{n_i}$. Function $\phi(z)$ is also $C^2$, $\mu_z$-strongly convex with its minimizer at $\phi_z^\star$.*

We refer to the indicators as block-separable because they take the form $\sum_i I_i(x_i)$ instead of $I(x_1, ..., x_n)$. The key distinction is that, in the block-separable case, each block of $x$ must belong to a specific convex and closed set. Note that this is not a restrictive assumption for a wide range of problems in robotics, optimal control, and related areas, as the objective functions in these applications typically represent quadratic costs, often combined with indicator functions to enforce safe regimes for the control variables $x$. Moreover, separable non-smooth functions have been frequently assumed in numerous works such as Lin et al. (2016); Deng et al. (2017); Yang et al. (2022).

**Definition 2.4.** *Function $g(\cdot) : \mathbb{R}^m \to \mathbb{R}$ is called $L$-smooth when there exists $L > 0$ such that*

$$\|\nabla g(x) - \nabla g(x')\| \leq L\|x - x'\|, \ \forall x, x' \in \mathbb{R}^m.$$

Note that the indicator functions $\{I_i(\cdot)\}$ may not be smooth.

**Definition 2.5.** *Function $g(\cdot) : \mathbb{R}^m \to \mathbb{R} \cup \{\infty\}$ is said to have the $\alpha$-PL property, where $\alpha \in (1, 2]$, if there exist $\eta \in (0, \infty]$ and $C > 0$, such that for all $x$ with $|g(x) - g(x^\star)| \leq \eta$, where $x^*$ is a point for which $\mathbf{0} \in \partial g(x^\star)$, we have*

$$\left(\text{dist}(\partial g(x), 0)\right)^\alpha \geq C|g(x) - g(x^\star)|.$$

It is important to note that when a function is subanalytic and lower semi-continuous then there exists an $\alpha$ such that it has also the $\alpha$-PL property, which is well-known in the non-convex optimization literature Frankel et al. (2015); Bento et al. (2024). Furthermore, it has been shown by Fatkhullin et al. (2022); Li et al. (2023) that under some mild conditions, the $\alpha$-PL property guarantees that the iterates of gradient-based algorithms such as gradient descent (GD) or stochastic GD converge to the optimizer with an explicit convergence rate. In the next section, by showing the $\alpha$-PL property for different scenarios, we could establish the convergence rate of the ADMM.

**Assumption 2.6.** *Matrix $Q \in \mathbb{R}^{n_c \times n_z}$ is full row rank.*

This assumption is weaker than the one made in Nishihara et al. (2015); Deng et al. (2017), where $Q$ is required to have full column rank. The full column rank can be replaced by a non-singularity assumption as one can reform the constraints to $Q^T A(x) + Q^T Q z = 0$ in which $Q^T Q$ is non-singular when $Q$ is full column rank. However, this reformulation is not possible when $Q$ is full row rank. Assumption 2.6 is crucial for establishing the convergence of ADMM, as demonstrated by the example below, where its violation leads to failure of the algorithm.

**Algorithm:** To solve equation 1, we consider the augmented Lagrangian ADMM, introducing a dual variable $w \in \mathbb{R}^{n_c}$ and a quadratic penalty term for the constraints with the coefficient $\rho$. This results in the following Lagrangian function.

**Definition 2.7.** *The corresponding augmented Lagrangian of equation 1 is given by*

$$L(x, z, w) := F(x) + \phi(z) + \langle w, A(x) + Qz \rangle + \frac{\rho}{2} \|A(x) + Qz\|^2, \tag{3}$$

*where $\rho > 0$ is the penalty parameter.*

ADMM is a powerful algorithm that can iteratively find a stationary point of the above Lagrangian function. Algorithm 1 summarizes the steps of this algorithm. At each iteration, it sequentially updates the current estimate by minimizing the augmented Lagrangian with respect to the variables in block $x_i$, $i \in \{1, ..., n\}$, and $z$ when all other blocks are fixed at their current estimates. Afterwards, the dual variable $w$ is updated depending on how much the constraints are violated. It is important to note

---

**Algorithm 1** ADMM

**Require:** $(x_1^0, \ldots, x_n^0), z^0, w^0, \rho$
    **for** $k = 0, 1, 2, \ldots$ **do**
        **for** $i = 1, \ldots, n$ **do**
            $x_i^{k+1} \in \arg\min_{x_i} L(x_{1:i-1}^{k+1}, x_i, x_{i+1:n}^k, z^k, w^k)$
        **end for**
        $z^{k+1} \in \arg\min_z L(x^{k+1}, z, w^k)$
        $w^{k+1} = w^k + \rho(A(x^{k+1}) + Qz^{k+1})$
    **end for**

---

that the minimization sub-problems for updating each block are derived by fixing all other blocks. This results in an augmented Lagrangian corresponding to a linearly constrained problem, which is tractable and can be solved efficiently Lin et al. (2015b; 2018).

To show the importance of Assumption 2.6 consider the following example in which this assumption is violated while other aforementioned assumptions hold but the ADMM fails to converge to a feasible solution. Therefore, Assumption 2.6 is indispensable.

**Example 2.8.** *Gao et al. (2020) Consider the following problem which is of the form of equation 1,*

$$\min_{x,y} x^2 + y^2, \quad s.t. \quad xy = 1.$$

*With an arbitrary initial point of the form $(x^0, 0, w^0)$, the ADMM's iterates will satisfy $(x^k, y^k) \to (0, 0)$ and $w^k \to -\infty$. Note that the limit point violates the constraint.*

## 3 THEORETICAL RESULTS

Herein, we present our theoretical guarantees for ADMM when applied to Problem equation 1. In particular, the following theorem demonstrates that, under the stated assumptions, ADMM converges and establishes key properties of the limit point.

**Theorem 3.1.** *Suppose that Assumption 2.3 and Assumption 2.6 hold. If $\phi$ is $L_\phi$-smooth, then ADMM in 1 with $\rho \geq \max\{\frac{4L_\phi^2}{\mu_\phi \lambda_{\min}^+(Q^T Q)}, \frac{4L_\phi^2}{\mu_\phi \sqrt{\lambda_{\min}^+(QQ^T)}}\}$ converges with at least sublinear convergence rate to a stationary point $(x^\star, z^\star, w^\star)$ of the augmented Lagrangian, i.e.,*

$$L(x^k, z^k, w^k) - L(x^\star, z^\star, w^\star) \in o(1/k),$$

*where $x^\star = (x_1^\star, ..., x_n^\star)$ and for all $i$,*

$$x_i^\star \in \arg\min_{x_i \in \mathbb{R}^{n_i}} f(x_i, x_{-i}^\star) + I_i(x_i), \quad s.t. \quad A(x_i, x_{-i}^\star) + Qz^\star = 0,$$

*and $z^\star$ is given by $z^\star \in \arg\min_{z \in \mathbb{R}^{n_z}} \phi(z)$, such that $A(x^\star) + Qz = 0$.*

This result shows that the limit point $(x^\star, z^\star)$ satisfies properties analogous to those of a Nash equilibrium point: that is, when all blocks except one are fixed at their limit values (e.g., $(x^\star_{-i}, z^\star)$), the objective function is optimized with respect to the remaining block (e.g., $x_i$). Previously, Gao et al. (2020) showed the convergence of ADMM but without providing any convergence rate. In addition, their result requires stronger assumptions, such as $f$ needs to be independent of certain variables, which is not needed in Theorem 3.1.

Next, we show that under additional assumptions such as the second-order differentiability of the Lagrangian at the limiting point and a sufficiently small degree of non-convexity, a linear convergence rate for ADMM when applied to equation 1 can be guaranteed. The degree of non-linearity is characterized by the relation between matrix $Q$ (the coefficient of the linear term in the constraints) and matrices $\{C_i\}$ (the coefficients of the non-linear term).

---

**Theorem 3.2.** *Suppose that the assumptions of Theorem 3.1 hold. Moreover, let $f$ be $L_f$-smooth and $L(x, z, w)$ is second-order differentiable at the limit point $(x^\star, z^\star, w^\star)$. When matrix $Q$ satisfies*

$$\|C\| \in \mathcal{O}\Big(\|(QQ^T)^{-1}Q\|^{-1} \cdot \min\Big\{m_1, m_2(\lambda_{min}(QQ^T))^{\frac{1}{2}}, m_3(\lambda_{min}(QQ^T))^{\frac{1}{4}}\|(QQ^T)^{-1}Q\|^{\frac{1}{2}}\Big\}\Big),$$
(4)

*where $\|C\| := \max_i \|C_i\|$ and constants $\{m_i \geq 0\}$ depending on problem's parameters e.g., $L_f, \mu_f, ...$ then, there exists $c_1 > 1$ such that the iterates of Algorithm 1 satisfy*

$$L(x^k, z^k, w^k) - L(x^\star, z^\star, w^\star) \in \mathcal{O}(c_1^{-k}).$$

*Furthermore, $(x^\star, z^\star)$ is a local minimum of the problem equation 1.*

---

Previous results by Lin et al. (2015b) on the performance of ADMM when applied to problems with linear constraints can be viewed as a special case of Theorem 3.2. Namely, when $\|C\| = 0$, the constraints in equation 1 reduce to linear constraints and subsequently, Equation (4) holds. Thus, according to the above result, the linear convergence of Lagrangian holds, which has been proved in the literature. In addition, Theorem 3.2 implies that even if nonlinear terms in the constraints exist, as long as $\|C\|$ is small enough, the linear convergence is still preserved.

It is important to emphasize that the above result requires differentiability of the Lagrangian at the limit point, which may not be valid in certain problems. Next, we replace this assumption with an additional minor assumption on the constraints for $x_i$s. Namely, we assume that they belong to some polyhedrals (see Appendix A). These types of constraints are common in various practical problems.

---

**Theorem 3.3.** *Under the assumptions of Theorem 3.1, when matrix $Q$ satisfies equation 4 and $\{I_i\}$ are the indicator functions of some polyhedral, then, the iterates of ADMM satisfy*

$$L(x^k, z^k, w^k) - \min_{(x,z) \in \mathcal{B}(x^k, z^k; r)} L(x, z, w^k) \in \mathcal{O}(c_2^{-k}),$$

*where $c_2 > 1$ and $r > 0$ are constant. Furthermore, $\lim_{k \to \infty}(x^k, z^k) = (x^\star, z^\star)$ is a local minimum of problem equation 1.*

---

**Approximated ADMM:** The algorithm in 1 is required to solve a series of sub-problems at each iteration in order to update $\{x_i\}$ and $z$. The results presented in the previous section are established under the assumption that these sub-problems are solved exactly. Although each sub-problem is strongly convex and efficiently solvable via gradient methods, it remains unclear whether the previous convergence rates hold under inexact solutions. In Appendix D.1, we introduce an approximated-ADMM and provide its convergence guarantees.

## 4 APPLICATION IN ROBOTICS

In the locomotion problem, the robot's centroidal momentum dynamics are considered (Viereck & Righetti, 2021; Meduri et al., 2023). The location, velocity, and angular momentum generated around the center of mass (CoM) are denoted by $\mathbf{c}$, $\dot{\mathbf{c}}$, and $\mathbf{k}$. We aim to optimize the objective function subject to the physics constraints, i.e., Newton-Euler equations. By discretizing the Newton-Euler equations and fixing the contact sequence and its timing, the optimal control problem for locomotion

can be written in the following unified way.

$$\min_{\mathbf{c},\dot{\mathbf{c}},\mathbf{k},\mathbf{f}} \quad \sum_{i=0}^{T-1} \phi_t(\mathbf{c}_i, \dot{\mathbf{c}}_i, \mathbf{k}_i, \mathbf{f}_i) + \phi_T(\mathbf{c}_T, \dot{\mathbf{c}}_T, \mathbf{k}_T), \tag{5}$$

$$\text{s.t.} \quad \mathbf{c}_{i+1} = \mathbf{c}_i + \dot{\mathbf{c}}_i \Delta t, \quad \dot{\mathbf{c}}_{i+1} = \dot{\mathbf{c}}_i + \sum_{j=1}^{N} \frac{\mathbf{f}_i^j}{m} \Delta t + \mathbf{g}\Delta t, \quad \dot{\mathbf{c}}_0 = \dot{\mathbf{c}}_{init}, \quad \mathbf{c}_0 = \mathbf{c}_{init},$$

$$\mathbf{k}_{i+1} = \mathbf{k}_i + \sum_{j=1}^{N}(\mathbf{r}_i^j - \mathbf{c}_i) \times \mathbf{f}_i^j \Delta t, \quad \mathbf{k}_0 = \mathbf{k}_{init}, \quad \mathbf{f}_i^j \in \Omega_i^j, \ \forall i,j,$$

where $\Delta t$ is the time discretization, subscript $i$ stands for time index, $T$ being the last one. Superscript $j$ specifies the index of the end-effector in contact with the environment, and $N$ is the number of the end-effector. Variables $\mathbf{c}_i, \dot{\mathbf{c}}_i, \mathbf{k}_i, \mathbf{f}_i^j$ denote the location, speed, angular momentum of the center of mass and the friction force at $j$-th contact at $i$-th discretization. The location of the end-effector in contact $\mathbf{r}$ is known. The initial conditions for the CoM are given by $\mathbf{c}_{init}, \dot{\mathbf{c}}_{init}, \mathbf{k}_{init}$. Function $\phi_t$ represents the running cost, $\phi_T$ is the terminal cost, and the friction $\mathbf{f}_i^j$ is constrained to lie within a safe region $\Omega_i^j$, which we assume it is cone and use polyhedral approximation to represent it. Note that this is a multi-affine equality constraint due to the term $\mathbf{c} \times \mathbf{f}$ in the angular momentum dynamics.

Next, we reformulate the problem into the form of equation 1 and apply the previous results to derive the ADMM convergence rate. First, notice that the variables $\mathbf{c}_i$ and $\dot{\mathbf{c}}_i$ can be rewritten as functions of $\mathbf{f} = \{\mathbf{f}_i\}$. See Appendix C for details. Second, by defining a new set of variables $\mathbf{k}' = \{\mathbf{k}'_i\}$ as $\mathbf{k}'_{i+1} := \mathbf{k}_{i+1} - \mathbf{k}_i$ for $i \geq 0$ and $\mathbf{k}'_0 := \mathbf{k}_{init}$ and assuming that the running and terminal costs can be decomposed into $f(\mathbf{f}) + \phi(\mathbf{k}')$, we obtain the following equivalent problem.

$$\min_{\mathbf{k}',\mathbf{f}} \quad f(\mathbf{f}) + \phi(\mathbf{k}') + \sum_{i=0}^{T} I_i(\mathbf{f}_i), \tag{6}$$

$$\text{s.t. } \mathbf{k}'_0 = \mathbf{k}_{init}, \ \mathbf{k}'_1 = \sum_{j=1}^{N}\Big((\mathbf{r}_0^j - \mathbf{c}_{init}) \times \mathbf{f}_0^j\Big)\Delta t, \ \mathbf{k}'_2 = \sum_{j=1}^{N}\Big((\mathbf{r}_1^j - \mathbf{c}_{init} - \dot{\mathbf{c}}_{init}\Delta t) \times \mathbf{f}_1^j\Big)\Delta t,$$

$$\mathbf{k}'_{i+1} = \sum_{j=1}^{N}\Big((\mathbf{r}_i^j - \mathbf{c}_{init} - \dot{\mathbf{c}}_{init}\, i(\Delta t) - \sum_{i'=0}^{i-2}(i-1-i')(\sum_{l=1}^{N}\frac{\mathbf{f}_{i'}^l}{m} + \mathbf{g})(\Delta t)^2) \times \mathbf{f}_i^j\Big)\Delta t, \ i \geq 2,$$

Problem in equation 6 has the same form as in equation 1. This can be seen by defining $z$ and $x$ in equation 1 to be $z := [\mathbf{k}'_0, \mathbf{k}'_1, \cdots, \mathbf{k}'_T]^T$ and $x := [x_0, \cdots, x_T]^T$, where $x_i := [\mathbf{f}_i^1, \cdots, \mathbf{f}_i^N]$. By denoting the corresponding Lagrangian function of the above problem as $L(\mathbf{f}, \mathbf{k}', w) = L(x, z, w)$, we can apply the results from the previous section.

**Corollary 4.1.** *Under the assumptions of Theorem 3.1 with sufficiently large $\rho$, the iterates of the ADMM applied to the problem in equation 6 satisfy $L(x^k, z^k, w^k) - L(x^\star, z^\star, w^\star) \in o(1/k)$.*

We also extend the result of Theorem 3.2 to the locomotion problem for which we require that the blocks of the initial point $x^0$, the global minimizer of $f(x)$ and $\phi(z)$, $x_f^\star$ and $\phi_z^\star$ are all bounded, i.e.,

$$\|x^0\|^2, \|x_f^\star\|^2 \in \mathcal{O}(n_x), \quad \|z_\phi^\star\|^2 \in \mathcal{O}(n_z). \tag{7}$$

This requirement holds in almost all physical problems. Note that equation 6 is a multi-affine quadratic constrained problem with the nonlinear term proportional to $(\Delta t)^3$. As $\Delta t \to 0$, the nonlinear term decays and subsequently, the linear convergence is guaranteed according to Theorem 3.2.

---

**Corollary 4.2.** *Under the assumptions of Corollary 4.1, if equation 7 holds and $L(x, z, w)$ is second-order differentiable at the limit point $(x^\star, z^\star, w^\star)$, then there exists $c_3 > 1$ and $t_0 > 0$, such that the iterates of the ADMM applied to the problem in equation 6 with $\Delta t \leq t_0$ satisfy*

$$L(x^k, z^k, w^k) - L(x^\star, z^\star, w^\star) \in \mathcal{O}(c_3^{-k}),$$

*Furthermore, $(x^\star, z^\star)$ is a local minimum of problem 6.*

---

In Appendix E, we further extend the result of Theorem 3.3 to the locomotion problem when $L$ is not second-order differentiable at $(x^\star, z^\star, w^\star)$ and show that linear convergence remains achievable.

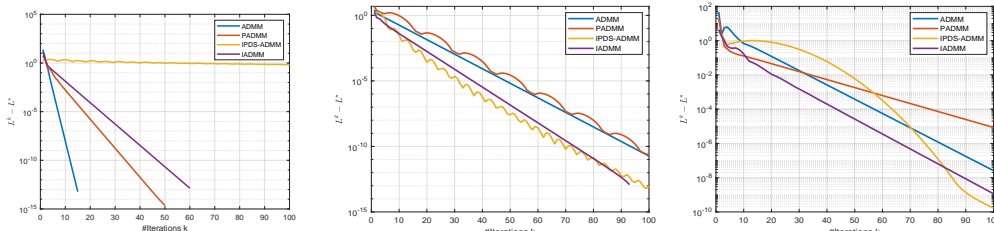

Figure 4: Left: convex objective with multi-affine constraint. Center: convex objective with linear constraint. Right: nonconvex objective with linear constraint

## 5 EXPERIMENTS

In this section, we first present a toy example to study the effect of the multi-affine constraint on the convergence rate of the ADMM. Next, we apply the ADMM algorithm to simplified 2D example of locomotion and dynamic locomotion.

**Effect of multi-affine quadratic constraint on the convergence rate:** Recall that the constraint set of the problem in (1) consists of two parts: the multi-affine quadratic operator $A(\cdot)$, and the linear part represented by $Q$. According to Theorems 3.2 and 3.3, when the linearity in the constraint becomes dominant, it results in a linear convergence rate.

To study the effect of linearity in the constraint on the ADMM's convergence, we consider the following,

$$\min_{\{x_i\},z} \frac{\mu_x}{2}(x_1^2 + x_2^2 + x_3^2 + x_4^2) + \frac{\mu_z}{2}z^2,$$
$$\text{s.t. } x_1 x_2 - x_3 x_4 + qz + 1 = 0.$$

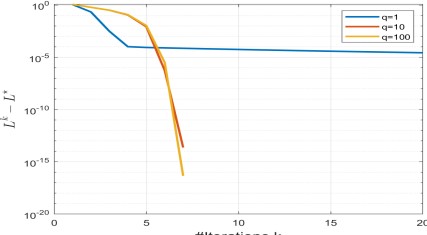

In this problem, the effect of non-linearity is quantified by the coefficient $q$. As $q$ becomes larger, it ensures the convergence rate is linear. Condition in (4) suggests $q \geq 10$ to ensure linear convergence. This is illustrated in Figure 2, showing the convergence results of ADMM under different $q$.

Figure 2: Performance of the ADMM under the effect of nonlinearity.

**Comparison with Existing Methods:** To further demonstrate the effectiveness of our method, we conduct a comparative study against PADMM of Yashtini (2021), IPDS-ADMM of Yuan (2025) and IADMM from Tang & Toh (2024) under three scenarios: (i) convex objective with multi-affine constraints, (ii) convex objective with linear constraints, and (iii) nonconvex objective with linear constraints. These methods aim to improve efficiency in linearly constrained problems. However, they currently lack theoretical guarantees when applied to nonlinear constrained problems. As illustrated in Figure 4, our algorithm achieves superior performance when the constraints are nonlinear, while comparable performance in other settings. This highlights the robustness and the efficiency of our approach beyond the convex setting.

**2D Locomotion problem:** Figure 5 depicts a 2D locomotion problem in which the goal is to achieve smooth walking behaviors, potentially involving varying step lengths at different time steps. To demonstrate the performance of the ADMM algorithm for finding the optimal trajectories, i.e., $\{\mathbf{f}_i, \mathbf{k}_i\}$, we considered this 2D version of the problem in (5).

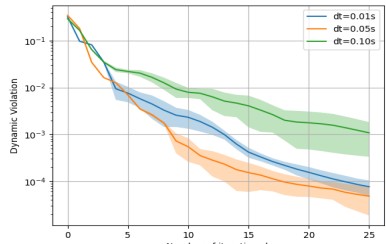

In this experiment, we selected a set of parameters that are close to a realistic application, namely, we set $m = 2$ kg (small-size robot in Grimminger et al. (2020)). We used the cost terms $f_i(\mathbf{f}_i) = \frac{1}{2}\sum_{i=0}^{T}\|\mathbf{f}_i\|^2 + I_i(\mathbf{f}_i)$, and $\phi(z) = 5\sum_{i=0}^{T}\|k_i'\|^2$. The constraints on $\mathbf{f}$ are designed to ensure that the center of mass remains within a specified target area.

Figure 3: Mean and standard deviation of dynamic violation values over optimization iterations. Results are shown for three different time discretizations. The x-axis shows the iteration number $k$.

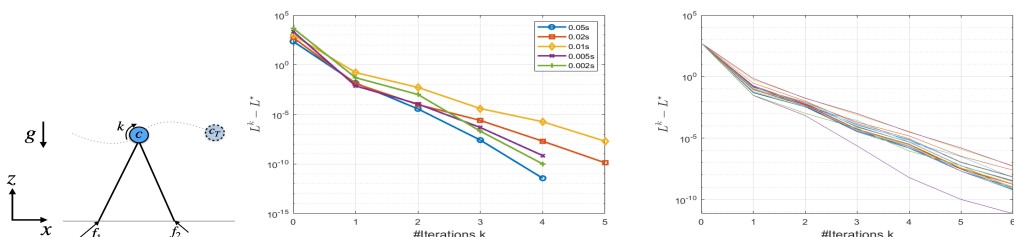

Figure 5: Left: Schematic of a 2D locomotion problem. The robot has two contacts with friction $f_1$ and $f_2$. The location and angular momentum are $\mathbf{c}$ and $\mathbf{k}$. Center: Performance of the ADMM for different $\Delta t$. Right: Convergence rate of the ADMM for the 2D problem with random initialization.

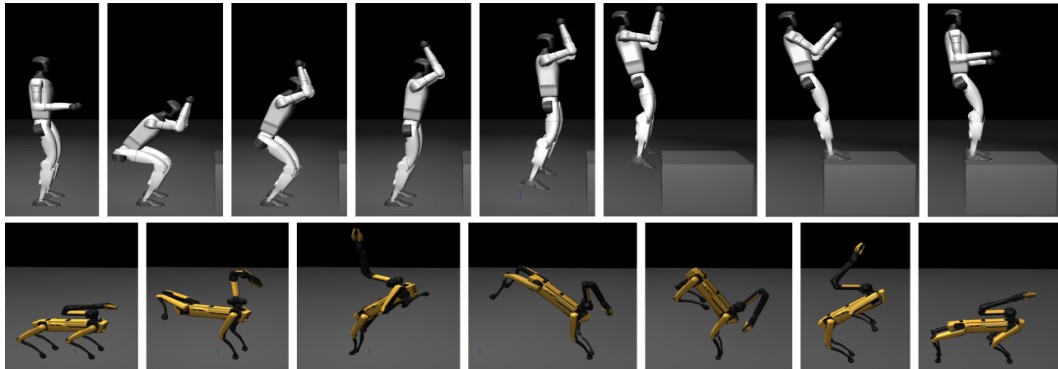

Figure 6: Snapshots of the robot experiments. The top row shows a humanoid robot performing a vertical jump. The bottom row illustrates a quadruped robot executing a bounding gait. In both cases, centroidal trajectories and forces are found using equation 5, and then a kinematic optimization tracks the planned centroidal trajectory.

Figure 5(center) illustrates the convergence rate for different discretization time values $\Delta t$. Note that the y-axis is on the log scale. As suggested by the result of Corollary 4.2, for small enough $\Delta t$, linear convergence is guaranteed by the ADMM. While $\Delta t = 0.005$ sec as suggested by Corollary 4.2, our empirical results in Fig. 5 indicate that the bound provided in the corollary is conservative. In practice, ADMM exhibits a linear convergence rate even for significantly larger values of $\Delta t$. As shown in Fig. 5(right), ADMM consistently converges linearly regardless of the initial configuration.

**Dynamic locomotion problem:** Figure 6 depicts dynamic motions executed on a humanoid and quadrupedal robot. These motions can be described by a fixed contact sequence and transition times, which can be used to formulate equation 5. The resulting CoM trajectory $(\mathbf{c}, \dot{\mathbf{c}}, \mathbf{k})$ can then be tracked via a kinematics optimization in order be applied on a robotic system as depicted in the figure.

In this experiment, we show successful transfer of the centroidal trajectories found using equation 5 or its equivalent in equation 6 via Algorithm 1 to high-dimensional robotics systems. The kinematics optimization is executed using an open source implementation of Differential Dynamic Programming (DDP) Mastalli et al. (2020). We report the centroidal dynamics constraint violation per iteration of Algorithm 1 for the jumping motion of the humanoid for three different discretization values $\Delta t$. The results are depicted in Figure 3, displaying the mean and standard deviation for each $\Delta t$ over 10 trials with randomized initial conditions.

## 6 CONCLUSION

In this paper, we provided theoretical guarantees for the convergence rate of ADMM when applied to a class of multi-affine quadratic equality-constrained problems. We proved that the sublinear convergence of the Lagrangian always holds, and every block of the limit point is the optimal solution when other blocks are fixed. We further proved that when the degree of non-convexity, measured by $\|C\|$, is small enough, the convergence will be linear. In addition, the limit point is a local minimum of the problem. Moreover, we applied our result to the locomotion problem in robotics. Our experimental results validated the correctness and robustness of our theorem. In the future, we plan to extend our results with higher-order non-linearity in the constraints and perform an extensive experiment on real-world applications.

## 7 REPRODUCIBILITY STATEMENT

The main paper specifies the problem formulation (Section 2) and theoretical guarantees (Section 3). Details of the algorithm, assumptions, and proofs are provided in the appendix. The robotics application (Section 4) and experiments (Section 5) are described with sufficient information for implementation, and additional details are included in the appendix and the supplementary material.

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

## Appendix

## A  Technical Definitions and Lemmas

**Definition A.1.** *Let $f : \mathbb{R}^n \to \mathbb{R} \cup \{\infty\}$ and $x \in \mathrm{dom}(f)$. A vector $v$ is a regular subgradient of $f$ at $x$, indicated by $v \in \widehat{\partial} f(x)$, if $f(y) \geq f(x) + \langle v, y - x \rangle + o(\|y - x\|)$ for all $y \in \mathbb{R}^n$. A vector $v$ is a general subgradient, indicated by $v \in \partial f(x)$, if there exist sequences $x_n \to x$ and $v_n \to v$ with $f(x_n) \to f(x)$ and $v_n \in \widehat{\partial} f(x_n)$.*

**Definition A.2** (Subanalytic set). *A subset $V \subset \mathbb{R}^n$ is called subanalytic if for every point $x \in \mathbb{R}^n$ there exist*
*- an open neighborhood $U \subset \mathbb{R}^n$ of $x$,*
*- a real-analytic manifold $M$ of dimension $n + m$,*
*- a relatively compact semianalytic set $S \subset M$,*
*and a real-analytic projection map $\pi \colon M \to \mathbb{R}^n$ such that $V \cap U = \pi(S) \cap U$.*

**Definition A.3** (Subanalytic function). *Let $U \subset \mathbb{R}^n$ be open. An extended-real-valued function $f \colon U \longrightarrow \mathbb{R} \cup \{\pm\infty\}$ is called subanalytic if its graph $\Gamma_f = \{(x, y) \in U \times \mathbb{R} : y = f(x)\}$ is a subanalytic subset of $\mathbb{R}^{n+1}$.*

**Definition A.4.** *(KL property Boţ & Nguyen (2020)) The function $\Psi$ is said to have the Kurdyka–Łojasiewicz (KL) property at a point $\hat{u} \in \operatorname{dom} \partial \Psi := \{u \in \mathbb{R}^N : \partial \Psi(u) \neq \emptyset\}$, if there exist $\eta \in (0, +\infty]$, a neighborhood $U$ of $\hat{u}$ and a function $\varphi \in \Phi_\eta$ such that for every*

$$u \in U \cap [\, \Psi(\hat{u}) < \Psi(u) < \Psi(\hat{u}) + \eta \,]$$

*it holds*

$$\varphi'(\Psi(u) - \Psi(\hat{u})) \cdot \operatorname{dist}(0, \partial \Psi(u)) \geq 1,$$

*where $\Phi_\eta$ is the set of all concave and continuous functions $\varphi : [0, \eta) \to [0, +\infty)$ which satisfy the following conditions:*

*1. $\varphi(0) = 0$;*

*2. $\varphi$ is $C^1$ on $(0, \eta)$ and continuous at $0$;*

*3. for all $s \in (0, \eta): \varphi'(s) > 0$.*

*If $\phi(x) = Cx^{(\alpha-1)/\alpha}$ in the KL property, where $C$ is a positive constant, then it is equivalent with $\alpha$-PL condition in Definition 2.5.*

**Definition A.5.** *A polyhedral set is a set which can be expressed as the intersection of a finite set of closed half-spaces, i.e., $\{x \in \mathbb{R}^n | Ax \leq b\}$ as for some matrix $A \in \mathbb{R}^{m \times n}$ and vector $b \in \mathbb{R}^m$.*

**Lemma A.6.** *Rockafellar & Wets (2009) If $f = g + f_0$ with $g$ finite at $\bar{x}$ and $f_0$ smooth on a neighborhood of $x$, then $\partial f(x) = \partial g(x) + \nabla f_0(x)$.*

**Lemma A.7.** *Rockafellar & Wets (2009) For any proper, convex function $f : \mathbb{R}^n \to \overline{\mathbb{R}}$ and any point $\bar{x} \in \operatorname{dom} f$, one has*

$$\partial f(\bar{x}) = \{v \mid f(x) \geq f(\bar{x}) + \langle v, x - \bar{x} \rangle \text{ for all } x\}.$$

**Lemma A.8** (Gao et al. (2020)). *Let $h$ be a $\mu$-strongly convex and $L$-smooth function, $A$ be a linear map of $x$, and $\mathcal{C}$ be a closed and convex set. Let $b_1, b_2 \in \operatorname{Im}(A)$, and consider the sets $\mathcal{U}_1 = \{x : Ax + b_1 \in \mathcal{C}\}$ and $\mathcal{U}_2 = \{x : Ax + b_2 \in \mathcal{C}\}$, which we assume to be nonempty. Let $x^* = \operatorname{argmin}\{h(x) : x \in \mathcal{U}_1\}$ and $y^* = \operatorname{argmin}\{h(y) : y \in \mathcal{U}_2\}$. Then, $\|x^* - y^*\| \leq \frac{1 + \frac{2L}{\mu}}{\sqrt{\lambda_{\min}^+(AA^T)}} \|b_2 - b_1\|$.*

**Theorem A.9** (Bolte et al. (2007)). *Assume that $f : \mathbb{R}^n \to \mathbb{R} \cup \{+\infty\}$ is a lower semi-continuous globally sub-analytic function and $f(x_0) = 0$, where $\mathbf{0} \in \partial f(x_0)$. Then, there exist $\delta > 0$ and $\theta \in [0, 1)$ such that for all $x \in |f|^{-1}(0, \delta)$, we have*

$$|f(x)|^\theta \leq \rho \|x^*\|, \quad \text{for all } x^* \in \partial f(x).$$

**Lemma A.10** (Inverse Function Theorem Clarke (1976)). *Let $u_0 \in X$ and $h_0 \in Y$ such that $g(u_0) = h_0$ and suppose that there exists a neighborhood $U_0 \subset X$ of $u_0$ such that 1) $g \in C^1$ for all the point in $U_0$; 2) $dg(u_0)$ is invertible. Then, there exist neighborhoods $U \subset U_0$ of $u_0$ and $V \subset Y$ of $h_0$, such that the equation $g(u) = h$ has a unique solution in $U$, for all $h \in V$.*

**Definition A.11** (Feehan (2019)). *Let $d \geq 1$ be an integer, $U \subset \mathbb{K}^d$ be an open subset, $E : U \to \mathbb{K}$ be a $C^2$ function, and $\operatorname{Crit}(E) := \{x \in U : \nabla E(x) = 0\}$. We say that $E$ is Morse-Bott at a point $x_\infty \in \operatorname{Crit}(E)$ if 1) $\operatorname{Crit}(E)$ is a $C^2$ sub-manifold of $U$, and 2) $T_{x_\infty} \operatorname{Crit}(E) = \operatorname{Ker} \nabla^2 E(x_\infty)$, where $T_x \operatorname{Crit}(E)$ is the tangent space to $\operatorname{Crit}(E)$ at a point $x \in \operatorname{Crit}(E)$.*

**Theorem A.12.** *Feehan (2019) Let $d \geq 1$ be an integer and $U \subset \mathbb{K}^d$ an open subset. If $E : U \to \mathbb{K}$ is a Morse-Bott function, then there are constants $C \in (0, \infty)$ and $\sigma_0 \in (0, 1]$ such that*

$$\|\nabla E(x)\| \geq C_0 |E(x) - E(x_\infty)|^{1/2}, \quad \text{for all } x \in \mathcal{B}(x_\infty; \sigma).$$

## B  ADDITIONAL EXPERIMENT DETAILS

The details of the problem in the comparison section are presented here. In that experiment, we consider the following problems and used three different optimizer including our ADMM and illustrated their convergence rates in Figure 4.

1. Convex objective with multi-affine constraints:

$$\min_{\{x_i\},z} \frac{\mu_x}{2}(x_1^2 + x_2^2 + x_3^2 + x_4^2) + \frac{\mu_z}{2}z^2, \quad \text{s.t.} \quad x_1 x_2 - x_3 x_4 + 1.5z + 1 = 0.$$

2. Convex objective with linear constraints:

$$\min_{\{x_i\},z} \frac{\mu_x}{2}(x_1^2 + x_2^2) + \frac{\mu_z}{2}z^2, \quad \text{s.t.} \quad x_1 + x_2 + z + 1 = 0.$$

3. Non-convex objective with linear constraints:

$$\min_{\{x_i\},z} \frac{\mu_x}{2}(x_1^2 + 4\sin^2(x_1) + x_2^2) + \frac{\mu_z}{2}z^2, \quad \text{s.t.} \quad x_1 + x_2 + z + 1 = 0.$$

## C DERIVATIONS OF THE LOCOMOTION PROBLEM

Notice that the variables $\mathbf{c}_i$ and $\dot{\mathbf{c}}_i$ for $i \geq 2$ in Problem Equation (5) can be rewritten as functions of $\mathbf{f} = \{\mathbf{f}_i\}$, as

$$\mathbf{c}_i(\mathbf{f}) = \mathbf{c}_{\text{init}} + \dot{\mathbf{c}}_{\text{init}}\, i(\Delta t) + \sum_{i'=0}^{i-2}(i - 1 - i')\Big(\sum_{j=1}^{N}\frac{\mathbf{f}_{i'}^j}{m} + \mathbf{g}\Big)(\Delta t)^2,$$

$$\dot{\mathbf{c}}_i(\mathbf{f}) = \dot{\mathbf{c}}_{init} + \sum_{i'=0}^{i-1}\sum_{j=1}^{N}\Big(\frac{\mathbf{f}_{i'}^j}{m} + \mathbf{g}\Big)(\Delta t).$$

## D PROOFS OF THEOREMS IN SECTION 3

PROOF OF THEOREM 3.1

The proof consists of three main parts: i) to show that $\{L(x^k, z^k, w^k)\}$ is decreasing, ii) to show that the sequence $\{x^k, z^k, w^k\}$ is bounded and has a limit point, and iii) to use the $\alpha$-PL property for establishing the convergence rate.

**i)** $L(x^k, z^k, w^k)$ **is decreasing:** From Assumption 2.3, $f$ is strongly convex for each blocks $i$, we get

$$f(x_{1:i-1}^{k+1}, x_{i:n}^k) - f(x_{1:i}^{k+1}, x_{i+1:n}^k) \geq \langle \nabla_i f(x_{1:i}^{k+1}, x_{i+1:n}^k), x_i^{k+1} - x_i^k \rangle + \frac{\mu_f}{2}\|x_i^{k+1} - x_i^k\|^2. \quad (8)$$

Furthermore, since $I_i$ is convex, Lemma A.7 implies

$$I_i(x_i^{k+1}) \geq I_i(x_i^k) + \langle v, x_i^{k+1} - x_i^k \rangle, \quad (9)$$

for all $v \in \partial I_i(x_i^t)$. As $x_i^{k+1} \in \text{argmin} L(x_{1:i-1}^{k+1}, x_i, x_{i+1:n}^k, z^k, w^k)$, the subgradient at $x_i^{k+1}$ satisfies

$$\mathbf{0} \in \partial L(x_{1:i}^{k+1}, x_{i+1:n}^k, z^k, w^k),$$
$$\Longrightarrow \mathbf{0} \in \partial I(x_i^{k+1}) + \nabla_i f(x_{1:i}^{k+1}, x_{i+1:n}^k) + \nabla_i A(x_{1:i}^{k+1}, x_{i+1:n}^k)w^k$$
$$+ \rho \nabla_i A(x_{1:i}^{k+1}, x_{i+1:n}^k)(A(x_{1:i}^{k+1}, x_{i+1:n}^k) + Qz^k),$$
$$\Longrightarrow -\nabla_i f(x_{1:i}^{k+1}, x_{i+1:n}^k) - \nabla_i A(x_{1:i}^{k+1}, x_{i+1:n}^k)(w^k + \rho(A(x_{1:i}^{k+1}, x_{i+1:n}^k) + Qz^k)) \in \partial I(x_i^{k+1}).$$
$$(10)$$

where the second line holds from the fact that $f(x) + \langle w, A(x) + Qz \rangle + \frac{\rho}{2}\|A(x) + Qz\|^2$ is first-order differentiable and 8.8(c) of Rockafellar & Wets (2009). On the other hand, we have

$$L(x_{1:i-1}^{k+1}, x_{i:n}^k, z^k, w^k) - L(x_{1:i}^{k+1}, x_{i+1:n}^k, z^k, w^k)$$
$$= f(x_{1:i-1}^{k+1}, x_{i:n}^k) - f(x_{1:i}^{k+1}, x_{i+1:n}^k) + I_i(x_i^k) - I_i(x_i^{k+1}) + \langle w^k, A(x_{1:i-1}^{k+1}, x_{i:n}^k) - A(x_{1:i}^{k+1}, x_{i+1:n}^k) \rangle$$
$$+ \frac{\rho}{2}(\|A(x_{1:i-1}^{k+1}, x_{i:n}^k) + Qz^k\|^2 - \|A(x_{1:i}^{k+1}, x_{i+1:n}^k) + Qz^k\|^2)$$
$$= f(x_{1:i-1}^{k+1}, x_{i:n}^k) - f(x_{1:i}^{k+1}, x_{i+1:n}^k) + I_i(x_i^k) - I_i(x_i^{k+1}) + \langle w^k, A(x_{1:i-1}^{k+1}, x_{i:n}^k) - A(x_{1:i}^{k+1}, x_{i+1:n}^k) \rangle$$
$$+ \frac{\rho}{2}\|A(x_{1:i-1}^{k+1}, x_{i:n}^k) - A(x_{1:i}^{k+1}, x_{i+1:n}^k)\|^2 + \rho\langle A(x_{1:i-1}^{k+1}, x_{i:n}^k) - A(x_{1:i}^{k+1}, x_{i+1:n}^k), A(x_{1:i}^{k+1}, x_{i+1:n}^k) + Qz^k \rangle$$

$$= f(x_{1:i-1}^{k+1}, x_{i:n}^k) - f(x_{1:i}^{k+1}, x_{i+1:n}^k) + I_i(x_i^k) - I_i(x_i^{k+1}) + \frac{\rho}{2}\|A(x_{1:i-1}^{k+1}, x_{i:n}^k) - A(x_{1:i}^{k+1}, x_{i+1:n}^k)\|^2,$$

$$+ \langle w^k + \rho(A(x_{1:i}^{k+1}, x_{i+1:n}^k) + Qz^k), A(x_{1:i-1}^{k+1}, x_{i:n}^k) - A(x_{1:i}^{k+1}, x_{i+1:n}^k)\rangle$$

$$= f(x_{1:i-1}^{k+1}, x_{i:n}^k) - f(x_{1:i}^{k+1}, x_{i+1:n}^k) + I_i(x_i^k) - I_i(x_i^{k+1}) + \frac{\rho}{2}\|A(x_{1:i-1}^{k+1}, x_{i:n}^k) - A(x_{1:i}^{k+1}, x_{i+1:n}^k)\|^2,$$

$$+ (x_i^k - x_i^{k+1})^T \nabla_i A(x_{1:i+1}^k, x_{i+1:n}^k)(w^k + \rho(A(x_{1:i}^{k+1}, x_{i+1:n}^k) + Qz^k))$$

$$= f(x_{1:i-1}^{k+1}, x_{i:n}^k) - f(x_{1:i}^{k+1}, x_{i+1:n}^k) + I_i(x_i^k) - I_i(x_i^{k+1}) + \frac{\rho}{2}\|A(x_{1:i-1}^{k+1}, x_{i:n}^k) - A(x_{1:i}^{k+1}, x_{i+1:n}^k)\|^2,$$

$$+ (x_i^k - x_i^{k+1})^T(-v - \nabla_i f(x_{1:i}^{k+1}, x_{i+1:n}^k))$$

$$= f(x_{1:i-1}^{k+1}, x_{i:n}^k) - f(x_{1:i}^{k+1}, x_{i+1:n}^k) - \langle \nabla_i f(x_{1:i}^{k+1}, x_{i+1:n}^k), x_i^k - x_i^{k+1}\rangle$$

$$+ I_i(x_i^k) - I_i(x_i^{k+1}) - \langle v, x_i^k - x_i^{k+1}\rangle + \frac{\rho}{2}\|A(x_{1:i}^{k+1}, x_{i+1:n}^k) - A(x_{1:i-1}^{k+1}, x_{i:n}^k)\|^2.$$

where $v \in \partial I_i(x_i^{k+1})$. The second equality is due to $\|V_1 - V_3\|^2 - \|V_2 - V_3\|^2 = \|V_2 - V_3\|^2 + 2\langle V_1 - V_2, V_2 - V_3\rangle$, for all the vectors $V_1$, $V_2$ and $V_3$. The fourth equality holds since $A(x_i, x_{-i})$ is an affine function for $x_i$ when $x_{-i}$ is fixed. We apply the Equation (10) at the fifth equality.

Consider the convexity property of $f$ and $I_i$ from Equation (8), Equation (9), the difference of Lagrangian can be further lower-bounded as

$$L(x_{1:i-1}^{k+1}, x_{i:n}^k, z^k, w^k) - L(x_{1:i}^{k+1}, x_{i+1:n}^k, z^k, w^k)$$

$$= f(x_{1:i-1}^{k+1}, x_{i:n}^k) - f(x_{1:i}^{k+1}, x_{i+1:n}^k) - \langle \nabla_i f(x_{1:i}^{k+1}, x_{i+1:n}^k), x_i^k - x_i^{k+1}\rangle$$

$$+ I_i(x_i^k) - I_i(x_i^{k+1}) - \langle v, x_i^k - x_i^{k+1}\rangle + \frac{\rho}{2}\|A(x_{1:i}^{k+1}, x_{i+1:n}^k) - A(x_{1:i-1}^{k+1}, x_{i:n}^k)\|^2$$

$$\geq f(x_{1:i-1}^{k+1}, x_{i:n}^k) - f(x_{1:i}^{k+1}, x_{i+1:n}^k) - \langle \nabla_i f(x_{1:i}^{k+1}, x_{i+1:n}^k), x_i^k - x_i^{k+1}\rangle$$

$$+ I_i(x_i^k) - I_i(x_i^{k+1}) - \langle v, x_i^k - x_i^{k+1}\rangle$$

$$\geq \frac{\mu_f}{2}\|x_i^{k+1} - x_i^k\|^2.$$

Adding up the above inequality for all the blocks, the difference of Lagrangian between $(x^k, z^k, w^k)$ and $(x^{k+1}, z^k, w^k)$ can be lower-bounded,

$$L(x^k, z^k, w^k) - L(x^{k+1}, z^k, w^k) \geq \frac{\mu_f}{2}\|x^{k+1} - x^k\|^2. \tag{11}$$

From the strong convexity of $\phi(z)$, the partial Hessian of Lagrangian on variable $z$ is

$$\nabla_{zz}^2 L(x, z, w) = \nabla_{zz}^2 \phi(z) + \rho Q^T Q \succeq \mu_\phi I,$$

which indicates the Lagrangian is $\mu_\phi$-strongly convex at $z$. As $z^{k+1} \in \arg\min_z L(x^{k+1}, z, w^k)$ and the strong-convexity of Lagrangian at $z$, the difference of Lagrangian between $(x^{k+1}, z^k, w^k)$ and $(x^{k+1}, z^{k+1}, w^k)$ can be lower-bounded,

$$L(x^{k+1}, z^k, w^k) - L(x^{k+1}, z^{k+1}, w^k) \geq \frac{\mu_\phi}{2}\|z^k - z^{k+1}\|^2. \tag{12}$$

From the update rule on $z$ and $w$, the partial derivative on $z$ at $(x^{k+1}, z^{k+1}, w^k)$ satisfies,

$$0 = \nabla_z L(x^{k+1}, z^{k+1}, w^k) = \nabla\phi(z^{k+1}) + Q^T w^k + \rho Q^T(A(x^k+1) + Qz^{k+1}) = \nabla\phi(z^{k+1}) + Q^T w^{k+1}.$$

which indicates $Q^T w^k = -\nabla\phi(z^k)$ if $k \geq 1$. In addition, from the setting that $Q^T w^0 = -\nabla\phi(z^0)$, we have $Q^T w^k = -\nabla\phi(z^k)$ for all $k \geq 0$. Therefore, for all $k$, we get

$$\|w^{k+1} - w^k\|^2 \leq \frac{\|Q^T w^{k+1} - Q^T w^k\|^2}{\lambda_{min}^+(Q^T Q)} = \frac{\|\nabla\phi(z^{k+1}) - \nabla\phi(z^k)\|^2}{\lambda_{min}^+(Q^T Q)} \leq \frac{L_\phi^2 \|z^{k+1} - z^k\|^2}{\lambda_{min}^+(Q^T Q)}. \tag{13}$$

After updating $w$, the Lagrangian between $(x^{k+1}, z^{k+1}, w^k)$ and $(x^{k+1}, z^{k+1}, w^{k+1})$ is lower-bounded by

$$L(x^{k+1}, z^{k+1}, w^k) - L(x^{k+1}, z^{k+1}, w^{k+1}) = \langle w^k - w^{k+1}, A(x^{k+1}) + Qz^{k+1}\rangle$$

$$= -\frac{1}{\rho}\|w^{k+1} - w^k\|^2 \geq -\frac{L_\phi^2 \|z^{k+1} - z^k\|^2}{\rho\lambda_{min}^+(Q^T Q)}. \tag{14}$$

where the inequality is due to Equation (13). As a result, from Equation (11), Equation (12) and Equation (47), the difference of Lagrangian between $(x^k, z^k, w^k)$ and $(x^{k+1}, z^{k+1}, w^{k+1})$ satisfies

$$
\begin{aligned}
L(x^k, z^k, w^k) &- L(x^{k+1}, z^{k+1}, w^{k+1}) \\
&\geq \frac{\mu_f}{2}\|x^{k+1} - x^k\|^2 + (\frac{\mu_\phi}{2} - \frac{L_\phi^2}{\rho \lambda_{min}^+(Q^T Q)})\|z^{k+1} - z^k\|^2, \\
&\geq \frac{\mu_f}{2}\|x^{k+1} - x^k\|^2 + \frac{\mu_\phi}{4}\|z^{k+1} - z^k\|^2. \\
&\geq \frac{\mu_f}{2}\|x^{k+1} - x^k\|^2 + \frac{\mu_\phi}{8}\|z^{k+1} - z^k\|^2 + \frac{\mu_\phi \lambda_{min}^+(Q^T Q)}{8 L_\phi^2}\|w^{k+1} - w^k\|^2, \forall k.
\end{aligned}
\tag{15}
$$

where we apply $\rho \geq \frac{4 L_\phi^2}{\mu_\phi \lambda_{min}^+(Q^T Q)}$ in the second inequality. In consequence, $\{L_k\}_{k=0}^{+\infty}$ is decreasing.

**ii)** $(x^k, z^k, w^k)$ **is bounded:** For these iterate, by denoting $\tilde{z}^k \in \arg\min\{\phi(z)|A(x^k) + Qz = 0\}$, we have

$$
\begin{aligned}
L(x^k, z^k, w^k) &= f(x^k) + \phi(z^k) + \langle w^k, A(x^k) + Qz^k \rangle + \frac{\rho}{2}\|A(x^k) + Qz^k\|^2, \\
&= f(x^k) + \phi(z^k) + \langle w^k, Q(z^k - \tilde{z}^k) \rangle + \frac{\rho}{2}\|A(x^k) + Qz^k\|^2, \\
&= f(x^k) + \phi(z^k) + \langle \nabla\phi(z^k), \tilde{z}^k - z^k \rangle + \frac{\rho}{2}\|A(x^k) + Qz^k\|^2, \\
&= f(x^k) + \phi(\tilde{z}^k) + \phi(z^k) - \phi(\tilde{z}^k) + \langle \nabla\phi(z^k), \tilde{z}^k - z^k \rangle + \frac{\rho}{2}\|A(x^k) + Qz^k\|^2, \\
&\geq f(x^k) + \phi(\tilde{z}^k) - \frac{L_\phi}{2}\|z^k - \tilde{z}^k\|^2 + \frac{\rho}{2}\|A(x^k) + Qz^k\|^2.
\end{aligned}
\tag{16}
$$

where the third line comes from $\nabla\phi(z^k) + Q^T w^k = 0$, the fourth line is due to the Lipschitz gradient of $\phi$. Now, consider any $z'$ such that $-Qz^k + Qz' = 0$, then

$$
L(x^{k+1}, z', w^k) - L(x^{k+1}, z^k, w^k) = \phi(z') - \phi(z^k).
$$

Since $z^k \in \arg\min L(x^{k+1}, z, w^k)$, from the equation above, we get $\phi(z^k) \leq \phi(z')$. In words, $z^k \in \arg\min\{\phi(z)| -Qz^k + Qz = 0\}, \forall k \geq 1$. Notice that $\tilde{z}^k \in \arg\min\{\phi(z)|A(x^k) + Qz = 0\}$ and $\phi(z)$ is strongly convex and smooth, from Lemma A.8, Equation (16) can be further lower-bounded when $\rho$ is large enough,

$$
\begin{aligned}
L(x^k, z^k, w^k) &\geq f(x^k) + \phi(\tilde{z}^k) - \frac{L_\phi}{2}\|z^k - \tilde{z}^k\|^2 + \frac{\rho}{2}\|Q(z^k - \tilde{z}^k)\|^2, \\
&\geq f(x^k) + \phi(\tilde{z}^k) + (\frac{\rho}{2} - \frac{\gamma L_\phi}{2})\|A(x^k) + Qz^k\|^2, \\
&\geq f(x^k) + \phi(\tilde{z}^k), \\
&\geq f(x_f^\star) + \phi(z_\phi^\star) + \frac{\mu_f}{2}\|x^k - x_f^\star\|^2 + \frac{\mu_\phi}{2}\|\tilde{z}^k - z_\phi^\star\|^2.
\end{aligned}
\tag{17}
$$

where $\gamma = \frac{1 + \frac{2 L_\phi}{\mu_\phi}}{\sqrt{\lambda_{\min}^+(QQ^T)}}$ and we apply $\rho \geq \frac{4 L_\phi^2}{\mu_\phi \sqrt{\lambda_{\min}^+(QQ^T)}}$.

The last line of Equation (17) and $\{L(x^k, z^k, w^k)\}$ being decreasing indicate $\{x^k\}$ and $\{\tilde{z}^k\}$ are bounded. In addition, from the Equation (17), $\|\tilde{z}^k - z^k\|^2$ can be upper-bounded as

$$
\begin{aligned}
\|\tilde{z}^k - z^k\|^2 &\leq \frac{2}{\rho - \gamma L_\phi}(L(x^k, z^k, w^k) - f(x^k) - \phi(\tilde{z}^k)), \\
&\leq \frac{2}{\rho - \gamma L_\phi}(L(x^0, z^0, w^0) - f(x_f^\star) - \phi(z_\phi^\star)).
\end{aligned}
$$

This implies that $\{z^k\}$ is bounded sequence as well.

As $Q$ is full row rank, $QQ^T$ is positive definite matrix. From the update rules of $w$, we know that $Q^T w^k = -\nabla\phi(z^k)$. Thus,

$$
\begin{aligned}
\|w^k\| &= \|(QQ^T)^{-1}Q\nabla\phi(z^k)\| \\
&\leq \|(QQ^T)^{-1}Q\| \cdot \|\nabla\phi(z^k) - \nabla\phi(z_\phi^\star)\| \\
&\leq \|(QQ^T)^{-1}Q\| \cdot \|\nabla\phi(z^k) - \nabla\phi(z_\phi^\star)\| \leq L_\phi\|(QQ^T)^{-1}Q\| \cdot \|z^k - z_\phi^\star\|.
\end{aligned}
$$

As $\{z^k\}$ is bounded, $\{w^k\}$ will be bounded. Hence, $\{x^k, z^k, w^k\}$ is bounded and a limit point exists.

From the definition of $L(x^{k+1}, z^{k+1}, w^{k+1})$, its subgradient at $x_i$ in $(k+1)$-th iteration is

$$
\begin{aligned}
\partial_i L(x^{k+1}, z^{k+1}, w^{k+1}) &= \partial I_i(x_i^{k+1}) + \nabla_i f(x^{k+1}) + \nabla_i A(x^{k+1})w^{k+1} \\
&\quad + \rho\nabla_i A(x^{k+1})(A(x^{k+1}) + Qz^{k+1}).
\end{aligned}
$$

From Equation (10), we have $-\nabla_i f(x_{1:i}^{k+1}, x_{i+1:n}^k) - \nabla_i A(x_{1:i}^{k+1}, x_{i+1:n}^k)(w^k + \rho(A(x_{1:i}^{k+1}, x_{i+1:n}^k) + Qz^k)) \in \partial I(x_i^{k+1})$, thus, according to the above equation, $v_i^{k+1} \in \partial_i L(x^{k+1}, z^{k+1}, w^{k+1})$ can be written as follows.

$$
\begin{aligned}
v_i^{k+1} &= -\nabla_i f(x_{1:i}^{k+1}, x_{i+1:n}^k) - \nabla_i A(x_{1:i}^{k+1}, x_{i+1:n}^k)(w^k + \rho(A(x_{1:i}^{k+1}, x_{i+1:n}^k) + Qz^k)) \\
&\quad + \nabla_i f(x^{k+1}) + \nabla_i A(x^{k+1})w^{k+1} + \rho\nabla_i A(x^{k+1})(A(x^{k+1}) + Qz^{k+1})
\end{aligned}
$$

Using algebraic manipulations, we can obtain

$$
\begin{aligned}
v_i^{k+1} &= \nabla_i f(x^{k+1}) - \nabla_i f(x_{1:i}^{k+1}, x_{i+1:n}^k) + (\nabla_i A(x^{k+1}) - \nabla_i A(x_{1:i}^{k+1}, x_{i+1:n}^k))w^{k+1} \\
&\quad + \nabla_i A(x_{1:i}^{k+1}, x_{i+1:n}^k)(w^{k+1} - w^k) + \rho(\nabla_i A(x^{k+1}) - \nabla_i A(x_{1:i}^{k+1}, x_{i+1:n}^k))A(x^{k+1}) \\
&\quad + \rho\nabla_i A(x_{1:i}^{k+1}, x_{i+1:n}^k)(A(x^{k+1}) - A(x_{1:i}^{k+1}, x_{i+1:n}^k)) \\
&\quad + \rho(\nabla_i A(x^{k+1}) - \nabla_i A(x_{1:i}^{k+1}, x_{i+1:n}^k))Qz^{k+1} + \rho\nabla_i A(x_{1:i}^{k+1}, x_{i+1:n}^k)Q(z^{k+1} - z^k).
\end{aligned}
\tag{18}
$$

By applying Equation (15), and the fact that $\{L^k\}$ is lower bounded from Equation (17), the norm of the difference between two iterates converge to 0, i.e.

$$
\lim_{k\to+\infty}\|x^{k+1} - x^k\| = 0, \quad \lim_{k\to+\infty}\|z^{k+1} - z^k\| = 0, \quad \lim_{k\to+\infty}\|w^{k+1} - w^k\| = 0. \tag{19}
$$

Because $\{x^k\}$ is bounded and $\{\nabla_i A(x^k)\}$ is multi-affine with respect to $x^k$, then $\{\nabla_i A(x^k)\}$ is bounded as well. From Equation (19), the limit of $\{v_i^k\}$ goes to the zero vector for all the blocks $i$, i.e.,

$$
\lim_{k\to+\infty} v_i^k = \mathbf{0} \in \partial_i L(x^k, z^k, w^k).
$$

For the variables $w$ and $z$, applying the update rule indicates

$$
\nabla_w L(x^{k+1}, z^{k+1}, w^{k+1}) = A(x^{k+1}) + Qz^{k+1} = \frac{1}{\rho}(w^{k+1} - w^k),
$$

and

$$
\begin{aligned}
\nabla_z L(x^{k+1}, z^{k+1}, w^{k+1}) &= \nabla\phi(z^{k+1}) + Q^T w^{k+1} + \rho Q^T(A(x^{k+1}) + Qz^{k+1}), \\
&= \rho Q^T(A(x^{k+1}) + Qz^{k+1}) = Q^T(w^{k+1} - w^k).
\end{aligned}
$$

Therefore, by applying Equation (19), the limit of the partial gradient $\lim_{k\to+\infty} \nabla_z L(x^k, z^k, w^k) = \mathbf{0}$ and $\lim_{k\to+\infty} \nabla_w L(x^k, z^k, w^k) = \mathbf{0}$. Overall, there exists $v^k \in \partial L(x^k, z^k, w^k)$ such that $\lim_{k\to+\infty} v^k = \mathbf{0}$. As the limit point $(x^\star, z^\star, w^\star)$ exists and $\mathbf{0} \in \partial L(x^\star, z^\star, w^\star)$, then $(x^\star, z^\star, w^\star)$ is a constrained stationary point.

On the other hand, since, the following problems are convex with affine constraints on $x_i$ and $z$, respectively

$$
\min_{x_i}\{f(x_i, x_{-i}^\star) + I_i(x_i) : A(x_i, x_{-i}^\star) + Q(z^\star) = 0\},
$$

$$
\min_z\{\phi(z) : A(x^\star) + Qz = 0\},
$$

they both satisfy the strong duality condition and thus $(x^\star, z^\star, w^\star)$ is also the global optimum of these problems. This finishes the first part of the proof.

**iii) Establishing the convergence rate:** Note that $I_i(x_i)$ is an indicator function of a closed semi-algebraic set. Subsequently, $L$ is a lower semi-continuous and sub-analytic function. Thus, applying Theorem A.9 to the function $L(x, z, w) - L(x^\star, z^\star, w^\star)$ shows that there exits $1 < \alpha \le 2$ and $\eta > 0$ such that it satisfies the $\alpha$-PL property,

$$(\text{dist}(0, \partial L(x, z, w)))^\alpha \ge c(L(x, z, w) - L(x^\star, z^\star, w^\star))$$

whenever $|L(x, z, w) - L(x^\star, z^\star, w^\star)| \le \eta$.

On the other hand, from Equation (15), there exists positive constants $a$ such that

$$L(x^{k+1}, z^{k+1}, w^{k+1}) - L(x^k, z^k, w^k) \le -a(\|x^{k+1} - x^k\|^2 + \|z^{k+1} - z^k\|^2 + \|w^{k+1} - w^k\|^2). \quad (20)$$

Moreover, from Equation (18) and the fact that $\{x_k, z_k, w_k\}$ is bounded, the norm of the subgradient is also upper-bounded the distance between two iterates, i.e.,

$$\|v^{k+1}\| \le b\sqrt{\|x^{k+1} - x^k\|^2 + \|z^{k+1} - z^k\|^2 + \|w^{k+1} - w^k\|^2}, \quad (21)$$

where $b > 0$. Therefore, by applying Equation (20) and Equation (21), the following inequality holds,

$$L(x^{k+1}, z^{k+1}, w^{k+1}) - L(x^k, z^k, w^k) \le -\frac{a}{b^2}\|v^{k+1}\|^2 \le -\frac{a}{b^2}\text{dist}\Big(0, \partial L(x^{k+1}, z^{k+1}, w^{k+1})\Big)^2. \quad (22)$$

From the $\alpha$-PL property, we obtain

$$L(x^{k+1}, z^{k+1}, w^{k+1}) - L(x^k, z^k, w^k) \le -\frac{ac^{2/\alpha}}{b^2}\Big(L(x^{k+1}, z^{k+1}, w^{k+1}) - L(x^\star, z^\star, w^\star)\Big)^{2/\alpha}.$$

whenever $|L(x^{k+1}, z^{k+1}, w^{k+1}) - L(x^\star, z^\star, w^\star)| \le \eta$. Then, from Theorem 1 and 2 of Bento et al. (2024), we have

$$L(x^k, z^k, w^k) - L(x^\star, z^\star, w^\star) \in \mathcal{O}(k^{-\frac{\alpha}{2-\alpha}}) \subseteq o\left(\frac{1}{k}\right).$$

Moreover, the iterates converges to the limit point, i.e., $\lim_{k \to +\infty}(x^k, z^k, w^k) = (x^\star, z^\star, w^\star)$. From Theorem 3 of Bento et al. (2024), the convergence rate of $(x^k, z^k, w^k)$ is

$$\|(x^k, z^k, w^k) - (x^\star, z^\star, w^\star)\| \in \mathcal{O}(k^{-\frac{\alpha-1}{2-\alpha}}).$$

$\square$

PROOF OF THEOREM 3.2

As Lagrangian is second-order differentiable at $(x^\star, z^\star, w^\star)$, we have $F(x) = f(x), \forall x \in \mathcal{B}(x^\star; r)$, where $r > 0$. In words, there exists a neighborhood of $(x^\star, z^\star, w^\star)$ which belongs to the constraint set, i.e., indicator functions are all zero for the points in that neighborhood. Recall the Lagrangian function,

$$L(x, z, w) = f(x) + \phi(z) + \sum_{i=1}^{n_c} w_i\Big(\frac{x^T C_i x}{2} + d_i^T x + e_i + q_i^T z\Big) + \frac{\rho}{2}\sum_{i=1}^{n_c}\Big(\frac{x^T C_i x}{2} + d_i^T x + e_i + q_i^T z\Big)^2.$$

Under Assumption 2.6, Assumption 2.3 and smoothness, we have $L_f I \succeq \nabla^2 f(x) \succeq \mu_f I$, $L_\phi I \succeq \nabla^2 \phi(z) \succeq \mu_\phi I$ and $Q^T = [q_1, \cdots, q_n]$ is full column rank. The constrained stationary point satisfies

$$\nabla_z L(x^\star, z^\star, w^\star) = \nabla\phi(z^\star) + Q^T w^\star = 0. \quad (23)$$

Next, we compute the Hessian of the Lagrangian at the stationary point $(x^\star, z^\star, w^\star)$ and show that if it is invertible then the convergence rate will be linear. To this end, applying Equation (23), the

Hessian at $(x^\star, z^\star, w^\star)$ can be represented as follows

$$
\begin{bmatrix}
\nabla^2 f(x^\star) + \sum_i w_i^\star C_i & \mathbf{0}_{n_x \times n_z} & C_1 x^\star + d_1 & \cdots & C_n x^\star + d_n \\
\mathbf{0}_{n_z \times n_x} & \nabla^2 \phi(z^\star) & q_1 & \cdots & q_n \\
(C_1 x^\star + d_1)^T & q_1^T & 0 & \cdots & 0 \\
\vdots & \vdots & \vdots & \ddots & \vdots \\
(C_n x^\star + d_n)^T & q_n^T & 0 & \cdots & 0
\end{bmatrix}
+ \rho \sum_{i=1}^{n_c} H_i,
$$

where

$$
H_i =
\begin{bmatrix}
C_i x^\star + d_i \\
q_i \\
0 \\
\vdots \\
0
\end{bmatrix}
\begin{bmatrix}
(C_i x^\star + d_i)^T & q_i^T & 0 & \cdots & 0
\end{bmatrix}.
$$

As a result, the kernel of the Hessian at $(x^\star, z^\star, w^\star)$ is equivalent to the kernel of the matrix in the first term. Which is invertible when the following matrix is invertible.

$$
\nabla^2 f(x^\star) + \sum_i w_i^\star C_i - \begin{bmatrix} \mathbf{0}_{n_x \times n_z} & C_1 x^\star + d_1, \cdots, C_n x^\star + d_n \end{bmatrix}
\begin{bmatrix} \nabla^2 \phi(z^\star) & Q^T \\ Q & \mathbf{0}_{n \times n} \end{bmatrix}^{-1}
\begin{bmatrix} \mathbf{0}_{n_z \times n_x} \\ (C_1 x^\star + d_1)^T \\ \vdots \\ (C_n x^\star + d_n)^T \end{bmatrix}.
$$

Note that $\begin{bmatrix} \nabla^2 \phi(z^\star) & Q^T \\ Q & \mathbf{0}_{n \times n} \end{bmatrix}^{-1}$ exists given that $Q$ is full row rank.

$$
\nabla^2 f(x^\star) + \sum_i w_i^\star C_i + \begin{bmatrix} C_1 x^\star + d_1, \cdots, C_n x^\star + d_n \end{bmatrix} (Q(\nabla^2 \phi(z^\star))^{-1} Q^T)^{-1}
\begin{bmatrix} (C_1 x^\star + d_1)^T \\ \vdots \\ (C_n x^\star + d_n)^T \end{bmatrix}. \quad (24)
$$

Without loss of generality, we can assume $I_i(x_i^0) = 0 \ \forall i$. As $L(x^t, z^t, w^t)$ is decreasing over $t$ if $\rho$ is large enough, by selecting $(z^0, w^0)$ such that $A(x^0) + Qz^0 = 0$ and $Q^T w^0 = -\nabla \phi(z^0)$, strong convexity and smoothness of $f$ and $\phi$ imply

$$
\begin{aligned}
& f(x_f^\star) + \phi(z_\phi^\star) + L_f(\|x_f^\star\|^2 + \|x^0\|^2) + L_\phi(\|z_\phi^\star\|^2 + \|z^0\|^2) \\
& \geq f(x_f^\star) + \phi(z_\phi^\star) + \frac{L_f}{2}\|x_f^\star - x^0\|^2 + \frac{L_\phi}{2}\|z_\phi^\star - z^0\|^2 \\
& \geq f(x^0) + \phi(z^0) = L(x^0, z^0, w^0) \\
& \geq L(x^\star, z^\star, w^\star) = f(x^\star) + \phi(z^\star) \\
& \geq f(x_f^\star) + \phi(z_\phi^\star) + \frac{\mu_f}{2}\|x^\star - x_f^\star\|^2 + \frac{\mu_\phi}{2}\|z^\star - z_\phi^\star\|^2.
\end{aligned}
$$

In the above expression, the first inequality is due to $a^2 + b^2 \geq (a-b)^2/2$, the second inequality is due to the smoothness of $f$ and $\phi$, the third inequality is because $L(x^k, z^k, w^k)$ is decreasing over $k$, and the last inequality is due to strong convexity of $f$ and $\phi$.

The above expression implies

$$
\frac{\mu_f}{2}\|x^\star - x_f^\star\|^2 + \frac{\mu_\phi}{2}\|z^\star - z_\phi^\star\|^2 \leq L_f(\|x_f^\star\|^2 + \|x^0\|^2) + L_\phi(\|z_\phi^\star\|^2 + \|z^0\|^2),
$$

which provides upper bounds for $\|x^\star - x_f^\star\|^2$ and $\|z^\star - z_\phi^\star\|^2$, i.e.,

$$
\|x^\star - x_f^\star\|^2 \in \mathcal{O}\left( \frac{L_f}{\mu_f}(\|x_f^\star\|^2 + \|x^0\|^2) + \frac{L_\phi}{\mu_f}(\|z_\phi^\star\|^2 + \|z^0\|^2) \right). \quad (25)
$$

and

$$
\|z^\star - z_\phi^\star\|^2 \in \mathcal{O}\left( \frac{L_f}{\mu_\phi}(\|x_f^\star\|^2 + \|x^0\|^2) + \frac{L_\phi}{\mu_\phi}(\|z_\phi^\star\|^2 + \|z^0\|^2) \right). \quad (26)
$$

They also give upper bound for $\|x^\star - x_f^\star\|$ and $\|z^\star - z_\phi^\star\|$, i.e.,

$$\|x^\star - x_f^\star\| \in \mathcal{O}\left(\sqrt{\frac{L_f}{\mu_f}}(\|x_f^\star\| + \|x^0\|) + \sqrt{\frac{L_\phi}{\mu_f}}(\|z_\phi^\star\| + \|z^0\|)\right), \qquad (27)$$

and

$$\|z^\star - z_\phi^\star\| \in \mathcal{O}\left(\sqrt{\frac{L_f}{\mu_\phi}}(\|x_f^\star\| + \|x^0\|) + \sqrt{\frac{L_\phi}{\mu_\phi}}(\|z_\phi^\star\| + \|z^0\|)\right). \qquad (28)$$

From the first Equation of Equation (23) and the fact that $Q$ is full row rank, an upper bound of $\|w^\star\|$ will be

$$\begin{aligned}
\|w^\star\| &= \|(QQ^T)^{-1}Q\nabla\phi(z^\star)\| \leq \|(QQ^T)^{-1}Q\| \cdot \|\phi(z^\star)\| \\
&= \|(QQ^T)^{-1}Q\| \cdot \|\phi(z^\star) - \phi(z_\phi^\star)\| \\
&\leq L_\phi \|(QQ^T)^{-1}Q\| \|z^\star - z_\phi^\star\| \\
&\in \mathcal{O}\left(\|(QQ^T)^{-1}Q\|L_\phi\left(\sqrt{\frac{L_f}{\mu_\phi}}(\|x_f^\star\| + \|x^0\|) + \sqrt{\frac{L_\phi}{\mu_\phi}}(\|z_\phi^\star\| + \|z^0\|)\right)\right).
\end{aligned} \qquad (29)$$

By the definition of $A(x)$, we have

$$\|A(x^0)\|^2 = \sum_{i=1}^{n_c}\left(\frac{1}{2}(x^0)^T C_i x^0 + d_i^T x^0 + e_i\right)^2 \in \mathcal{O}\left(n_c(\|x^0\|^4\|C\|^2 + \|x^0\|^2\|d\|^2 + \|e\|^2)\right),$$

where $\|C\| := \max_i \|C_i\|$, $\|d\| := \max_i \|d_i\|$ and $\|e\| := \max_i |e_i|$. Consequently,

$$\|A(x^0)\| \in \mathcal{O}\left(\sqrt{n_c}(\|x^0\|^2\|C\| + \|x^0\|\|d\| + \|e\|)\right). \qquad (30)$$

Notice that $A(x^0) + Qz^0 = 0$ and $Q$ is full row rank, then $z^0 = -Q^+ A(x^0) + v$, where $Q^+$ is the Moore-Penrose pseudo-inverse and $v \in \ker(Q)$. Thus, there exists $z^0$ such that

$$\|z^0\| = \|Q^+ A(x^0)\| \leq \|Q^+\|\|A(x^0)\| = \frac{1}{\sqrt{\lambda_{min}(QQ^T)}}\|A(x^0)\|.$$

From the bound of $\|A(x^0)\|$ in Equation (30), $\|z^0\|$ can be upper bounded as

$$\|z^0\| \leq \frac{1}{\sqrt{\lambda_{min}(QQ^T)}}\|A(x^0)\| \in \mathcal{O}\left(\sqrt{\frac{n_c}{\lambda_{min}(QQ^T)}}(\|x^0\|^2\|C\| + \|x^0\|\|d\| + \|e\|)\right). \qquad (31)$$

From Equation (29) and Equation (31), an upper bound of $\|w^\star\|$ can be rewritten as

$$\mathcal{O}\left(\|(QQ^T)^{-1}Q\|L_\phi\left(\sqrt{\frac{L_f}{\mu_\phi}}(\|x_f^\star\| + \|x^0\|) + \sqrt{\frac{L_\phi}{\mu_\phi}}\left(\|z_\phi^\star\| + \sqrt{\frac{n_c}{\lambda_{min}(QQ^T)}}(\|x^0\|^2\|C\| + \|x^0\|\|d\| + \|e\|)\right)\right)\right).$$

In addition, as $\|\sum_{i=1}^{n_c} w_i^\star C_i\| \in \mathcal{O}\left(\sqrt{n_c}\|w^\star\|\|C\|\right)$, it can be bounded as follows

$$\begin{aligned}
\mathcal{O}\Bigg(\|(QQ^T)^{-1}Q\|\sqrt{n_c}L_\phi\|C\|\Bigg(&\sqrt{\frac{L_f}{\mu_\phi}}(\|x_f^\star\| + \|x^0\|) \\
&+ \sqrt{\frac{L_\phi}{\mu_\phi}}\left(\|z_\phi^\star\| + \sqrt{\frac{n_c}{\lambda_{min}(QQ^T)}}(\|x^0\|^2\|C\| + \|x^0\|\|d\| + \|e\|)\right)\Bigg)\Bigg).
\end{aligned} \qquad (32)$$

Note that since $f$ is $\mu_f$-strongly convex, if $\|\sum_{i=1}^{n} w_i^\star C_i\| < \mu_f$, then the matrix in Equation (24) will be positive definite. So the Hessian at the stationary point is invertible. This can be ensured by letting each of the above terms to be $\mathcal{O}(\mu_f)$, i.e.,

$$\|(QQ^T)^{-1}Q\|\sqrt{n_c}L_\phi\|C\|\left(\sqrt{\frac{L_f}{\mu_\phi}}(\|x_f^\star\| + \|x^0\|) + \sqrt{\frac{L_\phi}{\mu_\phi}}\|z_\phi^\star\|\right) \in \mathcal{O}(\mu_f),$$

$$\|(QQ^T)^{-1}Q\|\|C\|\sqrt{\frac{L_\phi^3}{\mu_\phi}}\sqrt{\frac{n_c^2}{\lambda_{min}(QQ^T)}}(\|x^0\|\|d\| + \|e\|) \in \mathcal{O}(\mu_f), \qquad (33)$$

$$\|(QQ^T)^{-1}Q\|\|C\|^2\sqrt{\frac{L_\phi^3}{\mu_\phi}}\sqrt{\frac{n_c^2}{\lambda_{min}(QQ^T)}}\|x^0\|^2 \in \mathcal{O}(\mu_f).$$

The above inequalities lead to the following condition.

$$\|C\| \in \mathcal{O}\left(\min\left\{\frac{\mu_f\sqrt{\mu_\phi}}{\sqrt{n_c}L_\phi\left(\sqrt{L_f}(\|x_f^\star\| + \|x^0\|) + \sqrt{L_\phi}\|z_\phi^\star\|\right)}\|(QQ^T)^{-1}Q\|^{-1},\right.\right.$$

$$\frac{\mu_f\sqrt{\mu_\phi}}{\sqrt{L_\phi^3 n_c^2(\|x^0\|\|d\| + \|e\|)}}(\lambda_{min}(QQ^T))^{1/2}\|(QQ^T)^{-1}Q\|^{-1}, \tag{34}$$

$$\left.\left.\left(\frac{\mu_f\sqrt{\mu_\phi}}{\sqrt{L_\phi^3 n_c^2\|x^0\|^2}}\right)^{1/2}(\lambda_{min}(QQ^T))^{1/4}\|(QQ^T)^{-1}Q\|^{-1/2}\right\}\right).$$

By defining

$$m_1 := \frac{\mu_f\sqrt{\mu_\phi}}{\sqrt{n_c}L_\phi\left(\sqrt{L_f}(\|x_f^\star\| + \|x^0\|) + \sqrt{L_\phi}\|z_\phi^\star\|\right)},$$

$$m_2 := \frac{\mu_f\sqrt{\mu_\phi}}{\sqrt{L_\phi^3 n_c^2(\|x^0\|\|d\| + \|e\|)}},$$

$$m_3 := \left(\frac{\mu_f\sqrt{\mu_\phi}}{\sqrt{L_\phi^3 n_c^2\|x^0\|^2}}\right)^{1/2},$$

we obtain the condition in Equation (4).

If Equation (34) is satisfied, the stationary point is non-degenerate (meaning that the Hessian is invertible at that point). In consequence, the stationary point is also isolated. This can be seen by applying the Local Inversion Theorem presented in Lemma A.10 using $u_0 = (x^\star, z^\star, w^\star)$ and $g = \nabla L$. This lemma implies that every stationary point of the Lagrangian function is isolated if it is non-degenerate. As a result, if Equation (34) holds,

$$\ker\nabla^2 f(\tilde{x}) = T_{\tilde{x}}S = \{\mathbf{0}\},$$

where $\tilde{x}$ is a stationary point, $S$ is the set of stationary points, and $T_{\tilde{x}}S$ is the tangent space of $S$ at $\tilde{x}$. Therefore, according to Definition A.11 and Theorem A.12, the Lagrangian is a Morse-Bott function, and it satisfies the $\alpha$-PL inequality around every stationary point with $\alpha = 2$, i.e.,

**Theorem D.1.** *Suppose that the Equation* (4) *holds, then for every stationary point* $(x^\star, z^\star, w^\star)$ *of* $L$ *such that* $L$ *is second order differentiable at this point, there exists constants* $C$ *and* $r > 0$, *s.t.*

$$\|\nabla L(x, z, w)\|^2 \geq C|L(x, z, w) - L(x^\star, z^\star, w^\star)|, \quad \forall(x, z, w) \in \mathcal{B}(x^\star, z^\star, w^\star; r).$$

From Equation (22) and the above result, for $k$ large enough so that $(x^k, z^k, w^k) \in \mathcal{B}(x^\star, z^\star, w^\star; r)$, we have

$$L(x^{k+1}, z^{k+1}, w^{k+1}) - L(x^k, z^k, w^k) \leq -\frac{aC}{b^2}\left(L(x^{k+1}, z^{k+1}, w^{k+1}) - L(x^\star, z^\star, w^\star)\right),$$

$$\implies L(x^{k+1}, z^{k+1}, w^{k+1}) - L(x^\star, z^\star, w^\star) \leq \left(1 + \frac{aC}{b^2}\right)^{-1}\left(L(x^k, z^k, w^k) - L(x^\star, z^\star, w^\star)\right),$$

$$\implies L(x^k, z^k, w^k) - L(x^\star, z^\star, w^\star) = O(c^{-k}),$$

where $c := \left(1 + \frac{aC}{b^2}\right) > 1$. From Theorem 3 of Bento et al. (2024), the convergence of iterates $(x^k, z^k, w^k)$ is

$$\|x^k - x^\star\|^2 + \|z^k - z^\star\|^2 + \|w^k - w^\star\|^2 \in \mathcal{O}(c^{-k}).$$

Lastly, consider the second partial derivative of the Lagrangian with respect to $x$ and $z$ at $(x^\star, z^\star, w^\star)$,

$$\begin{bmatrix} \nabla^2 f(x^\star) + \sum_i w_i^\star C_i & \mathbf{0} \\ \mathbf{0} & \nabla^2\phi(z^\star) \end{bmatrix} \succ \mathbf{0}.$$

From proposition 3.3.2 of Bertsekas (1997), the second-order sufficiency condition is satisfied and thus, the point $(x^\star, z^\star)$ is the local minimum of the problem 1. □

As a corollary of Theorem 3.2, if the limiting point is of the form $(x^\star, z^\star, \mathbf{0})$, then the linear convergence rate of the ADMM is ensured, i.e.,

**Corollary D.2.** *Under the assumptions of Theorem 3.1, if the iterates of the ADMM converge to $(x^\star, z^\star, w^\star)$ with $w^\star = \mathbf{0}$, then, there is $c_1 > 1$ such that*

$$L(x^k, z^k, w^k) - L(x^\star, z^\star, w^\star) \in \mathcal{O}(c_1^{-k}).$$

*Proof.* When $w_i^\star = 0$, $\forall i$, the Equation Equation (24) will become

$$\nabla^2 f(x^\star) + \left[ C_1 x^\star + d_1, \cdots, C_n x^\star + d_n \right] (Q(\nabla^2 \phi(z^\star))^{-1} Q^T)^{-1} \begin{bmatrix} (C_1 x^\star + d_1)^T \\ \vdots \\ (C_n x^\star + d_n)^T \end{bmatrix} \succ 0.$$

and consequently, the Hessian of the Lagrangian at the stationary point will be invertible. The rest of the proof is similar to the proof of Theorem 3.2. □

PROOF OF THEOREM 3.3

Suppose that the constraint sets for $X_i$s are polyhedrals. From Equation (13), Equation (20) and Equation (21), there exist $v \in \partial_{x,z} L(x^{k+1}, z^{k+1}, w^{k+1})$, and positive constants $a$ and $b$, such that

$$L(x^{k+1}, z^{k+1}, w^{k+1}) - L(x^k, z^k, w^k) \le -a(\|x^{k+1} - x^k\|^2 + \|z^{k+1} - z^k\|^2),$$

$$\|v\|^2 \le b(\|x^{k+1} - x^k\|^2 + \|z^{k+1} - z^k\|^2).$$

This results in

$$L(x^{k+1}, z^{k+1}, w^{k+1}) - L(x^k, z^k, w^k) \le -\frac{a}{b}\|v\|^2 \le -\frac{a}{b} \min_{s \in \partial_{x,z} L(x^{k+1}, z^{k+1}, w^{k+1})} \|s\|^2. \tag{35}$$

Consider the function $\tilde{L}$ s.t. $L(x, z, w) = \tilde{L}(x, z, w) + \sum_{i=1}^n I_i(x_i)$. Then, when $w$ is fixed, the second-order derivative of $\tilde{L}$ is

$$\begin{bmatrix} \nabla^2 f(x^\star) + \sum_i w_i^\star C_i & \mathbf{0} \\ \mathbf{0} & \nabla^2 \phi(z^\star) \end{bmatrix}.$$

To ensure that at the constrained stationary point $(x^\star, z^\star, w^\star)$, the above matrix is positive definite, we require that $\nabla^2 f(x^\star) + \sum_{i=1}^n w_i^\star C_i \succ 0$. Notice that at the constrained stationary point, Equation (23) still holds. According to the proof of Theorem 3.2, when Equation (34) holds, then $\nabla^2 f(x^\star) + \sum_{i=1}^n w_i^\star C_i \succ 0$, and the Hessian of $\tilde{L}$ is positive definite. Consequently, the function $\tilde{L}(x, z, w)$ is locally strongly convex with respect to $(x, z)$, $\forall (x, z) \in \mathcal{B}(x^\star, z^\star; r')$ and $\forall w \in \mathcal{B}(w^\star, r'')$, for some $r' > 0$ and $r'' > 0$. Since $\mathrm{dom}(L)$ is polyhedral, and $L(x, z, w) = \tilde{L}(x, z, w) + \sum_{i=1}^n I_i(x_i)$, according to the results in Appendix F of Karimi et al. (2016), the $\alpha$-PL inequality holds for $\alpha = 2$, i.e.,

$$\min_{s \in \partial_{x,z} L(x,z,w)} \|s\|^2 \ge 2\mu \left( L(x, z, w) - \min_{(x,z) \in \mathcal{B}(x^\star, z^\star; r')} L(x, z, w) \right),$$

$$\forall (x, z) \in \mathcal{B}(x^\star, z^\star; r') \text{ and } \forall w \in \mathcal{B}(w^\star, r'').$$

On the other hand, since $(x^k, z^k, w^k) \to (x^\star, z^\star, w^\star)$, for $k$ that is large enough, i.e. $k \ge K$, there exists $r > 0$ such that $\mathcal{B}(x^k, z^k; r) \subseteq \mathcal{B}(x^K, z^K; 2r)$, $\mathcal{B}(x^k, z^k; 2r) \subseteq \mathcal{B}(x^\star, z^\star; r')$ and $w^k \in \mathcal{B}(w^\star, r'')$. As a result,

$$\min_{s \in \partial_{x,z} L(x^k, z^k, w)} \|s\|^2 \ge 2\mu(L(x^k, z^k, w) - \min_{(x,z) \in \mathcal{B}(x^\star, z^\star; r')} L(x, z, w)),$$

$$\ge 2\mu(L(x^k, z^k, w) - \min_{(x,z) \in \mathcal{B}(x^k, z^k; 2r)} L(x, z, w)), \quad k \ge K.$$

Combining the above inequality with Equation (35) yields that there exists $C_1 > 0$ such that for $k \geq K$,

$$L(x^{k+1}, z^{k+1}, w^{k+1}) - L(x^k, z^k, w^k) \leq -C_1\Big(L(x^{k+1}, z^{k+1}, w^{k+1}) - \min_{(x,z)\in\mathcal{B}(x^K,z^K;2r)} L(x, z, w^{k+1})\Big).$$

Note that due to the update rule in Algorithm 1, for every $x$ and $z$, we have $L(x, z, w^{k+1}) - L(x, z, w^k) = \rho\|A(x) + Qz\|^2 \geq 0$, and consequently,

$$\min_{(x,z)\in\mathcal{B}(x^K,z^K;2r)} L(x, z, w^{k+1}) \geq \min_{(x,z)\in\mathcal{B}(x^K,z^K;2r)} L(x, z, w^k).$$

As a result, for $k \geq K$, the following inequality, obtained from the previous two inequalities, holds

$$(1+C_1)\Big(L(x^{k+1}, z^{k+1}, w^{k+1}) - \min_{(x,z)\in\mathcal{B}(x^K,z^K;2r)} L(x, z, w^{k+1})\Big) \leq L(x^k, z^k, w^k) - \min_{(x,z)\in\mathcal{B}(x^K,z^K;2r)} L(x, z, w^k).$$

This implies

$$L(x^k, z^k, w^k) - \min_{(x,z)\in\mathcal{B}(x^K,z^K;2r)} L(x, z, w^k) \leq (1+C_1)^{-(k-K)}\Big(L(x^K, z^K, w^K) - \min_{(x,z)\in\mathcal{B}(x^K,z^K;2r)} L(x, z, w^K)\Big). \tag{36}$$

As $\mathcal{B}(x^k, z^k; r) \subseteq \mathcal{B}(x^K, z^K; 2r)$, from Equation (36), the linear convergence of $L(x^k, z^k, w^k) - \min_{(x,z)\in\mathcal{B}(x^k,z^k;r)} L(x, z, w^k)$ can be ensured, i.e., let $c_2 := 1 + C_1$, then

$$L(x^k, z^k, w^k) - \min_{(x,z)\in\mathcal{B}(x^k,z^k;r)} L(x, z, w^k) \leq L(x^k, z^k, w^k) - \min_{(x,z)\in\mathcal{B}(x^K,z^K;2r)} L(x, z, w^k),$$

$$\leq (1+C_1)^{-(k-K)}\Big(L(x^K, z^K, w^K) - \min_{(x,z)\in\mathcal{B}(x^K,z^K;2r)} L(x, z, w^K)\Big) \in \mathcal{O}(c_2^{-k}).$$

To prove that $(x^*, z^*)$ is a local minimum, notice that as $k$ goes to infinity, $(x^k, z^k, w^k) \to (x^\star, z^\star, w^\star)$, $I_i(x_i^\star) = 0, \forall i$ and Equation (36) implies that for all $(x, z) \in \mathcal{B}(x^\star, z^\star; r)$

$$F(x^\star) + \phi(z^\star) = f(x^\star) + \phi(z^\star) = L(x^\star, z^\star, w^\star) = \min_{(x,z)\in\mathcal{B}(x^\star,z^\star;r)} L(x, z, w^\star) \leq L(x, z, w^\star).$$

For all the points $(x, z) \in \mathcal{B}(x^\star, z^\star; r)$ satisfying $A(x) + Qz = 0$, $F(x) + \phi(z)$ can be bounded as

$$F(x^\star) + \phi(z^\star) = \min_{(x,z)\in\mathcal{B}(x^\star,z^\star;r)} L(x, z, w^\star) \leq L(x, z, w^\star) = F(x) + \phi(z).$$

And thus $(x^\star, z^\star)$ is a local minimum of problem Equation (1). $\qquad\square$

### D.1 APPROXIMATED ADMM

Consider the following algorithm.

---
**Algorithm 2** Approximated-ADMM

---
**Require:** $(x_1^0, \ldots, x_n^0), z^0, w^0, \rho$
  **for** $k = 0, 1, 2, \ldots$ **do**
    **for** $i = 1, \ldots, n$ **do**
      $x_i^{k+1} \approx \arg\min_{x_i} L(x_{1:i-1}^{k+1}, x_i, x_{i+1:n}^k, z^k, w^k)$
    **end for**
    $z^{k+1} \approx \arg\min_z L(x^{k+1}, z, w^k)$
    $w^{k+1} = w^k + \rho(A(x^{k+1}) + Qz^{k+1})$
  **end for**

---

**Theorem D.3.** *Under the assumptions of Theorem 3.2, if the Approximated ADMM in 2 is applied to Problem 1, and the following condition are satisfied,*
***P1:** the iterates $p^k := (x^k, z^k, w^k)$ are bounded, and $L(p^k) = L(x^k, z^k, w^k)$ is lower bounded,*
***P2:** there is a constant $C_1 > 0$ such that for all sufficiently large $k$,*

$$L(x^k, z^k, w^k) - L(x^{k+1}, z^{k+1}, w^{k+1}) \geq C_1\|p^{k+1} - p^k\|^2.$$

**P3:** *and there exists $d^{k+1} \in \partial L(x^{k+1}, z^{k+1}, w^{k+1})$ and $C_2 > 0$ such that for all sufficiently large $k$,*

$$\|d^{k+1}\| \le C_2 \|p^{k+1} - p^k\|.$$

*Then, the convergence results of Theorem 3.1, Theorem 3.2 and Theorem 3.3 remain valid under their respective additional assumptions.*

*Proof.* Remind that these presented conditions in this theorem are the key element to prove Theorem 3.1. In exact ADMM, the condition P2 and P3 are satisfied in Equation (15) and Equation (21). Conditions P2 and P3 imply that there exists $d^k \in \partial L(x^k, z^k, w^k)$ such that

$$L(x^{k+1}, z^{k+1}, w^{k+1}) - L(x^k, z^k, w^k) \le -C_1(\|x^{k+1} - x^k\|^2 + \|z^{k+1} - z^k\|^2 + \|w^{k+1} - w^k\|^2)$$

$$\le -\frac{C_1}{C_2^2} \|d^{k+1}\|^2 \le -\frac{C_1}{C_2^2}(dist(\mathbf{0}, \partial L(x^{k+1}, z^{k+1}, w^{k+1})))^2,$$

which is precisely the Equation (22) in the Proof of Theorem 3.1. From condition P1 and P2, the Lagrangian $L(x^k, z^k, w^k)$ is lower bounded and non-increasing, which indicates $L(p^k)$ converges as $k$ goes to infinity. Then, from P2, we know,

$$\|x^{k+1} - x^k\| \to 0, \quad \|z^{k+1} - z^k\| \to 0, \quad \|w^{k+1} - w^k\| \to 0.$$

Based on P1, the iterates $(x^k, z^k, w^k)$ are bounded, thus, the limit point exists. The rest of the proof is identical with the proof of Theorem 3.1. $\square$

**Error models.** Suppose there exist nonnegative sequences $\{\epsilon_k^{(i)}\}_{k \ge 0}$ and $\{\eta_k\}_{k \ge 0}$ such that

$$L(x_{1:i-1}^{k+1}, x_i^{k+1}, x_{i+1:n}^k, z^k, w^k) \le \min_{x_i} L(x_{1:i-1}^{k+1}, x_i, x_{i+1:n}^k, z^k, w^k) + \epsilon_k^{(i)}, \tag{37}$$

$$L(x^{k+1}, z^{k+1}, w^k) \le \min_z L(x^{k+1}, z, w^k) + \eta_k, \tag{38}$$

At iteration $k$, let $\widehat{x}_i^{k+1}$ and $\widehat{z}^{k+1}$ denote the exact minimizers of the corresponding block subproblems appearing in Algorithm 1 with the current arguments fixed; i.e.,

$$\widehat{x}_i^{k+1} \in \arg\min_{x_i} L(x_{1:i-1}^{k+1}, x_i, x_{i+1:n}^k, z^k, w^k),$$

$$\widehat{z}^{k+1} \in \arg\min_z L(x^{k+1}, z, w^k).$$

The iterates produced by Algorithm 2 are denoted by $x_i^{k+1}$ and $z^{k+1}$, with errors $e_{x,i}^{k+1} := x_i^{k+1} - \widehat{x}_i^{k+1}$ and $e_z^{k+1} := z^{k+1} - \widehat{z}^{k+1}$.

**Theorem D.4.** *Let Algorithm 2 produce $(x^k, z^k, w^k)$ and assume the inexactness condition from Equation (37) and Equation (38), then:*

1. *If $\sum_k (\sum_i \epsilon_k^{(i)} + \eta_k) < \infty$ and the assumptions for Theorem 3.1 hold, the sequence $\{L(x^k, z^k, w^k)\}$ converges and $\lim_{k \to +\infty} \|A(x^k) + Qz^k\| = 0$, $\lim_{k \to +\infty} \|x^{k+1} - x^k\| = 0$, $\lim_{k \to +\infty} \|z^{k+1} - z^k\| = 0$. Every limit point $(x^\star, z^\star, w^\star)$ is a stationary point of $L$, and $x^\star$ and $z^\star$ satisfy the blockwise optimality conditions stated in Theorem 3.1.*

2. *If further $\sum_i \epsilon_k^{(i)} + \eta_k \le C[\|x^{k+1} - x^k\|^2 + \|z^{k+1} - z^k\|^2 + \|w^{k+1} - w^k\|^2]$, where $C$ is a small enough constant, the convergence result of Theorem 3.1, Theorem 3.2 and Theorem 3.3 remain valid under their respective additional assumptions.*

*Proof.* We work under Assumption 2.3 (blockwise strong convexity of $f$ and $\varphi$), Assumption 2.6 (full row rank of $Q$), and the smoothness conditions used in Theorems 3.1 and 3.2. Let denote the inexact errors $e_{x,i}^{k+1} := x_i^{k+1} - \widehat{x}_i^{k+1}$, $e_z^{k+1} := z^{k+1} - \widehat{z}^{k+1}$.

By blockwise strong convexity, each univariate subproblem $\psi_i(x_i) := L(x_{1:i-1}^{k+1}, x_i, x_{i+1:n}^k, z^k, w^k)$ is $\mu_f$-strongly convex for some $\mu_f > 0$, hence quadratic growth yields

$$\psi_i(x_i^{k+1}) - \psi_i(\widehat{x}_i^{k+1}) \ge \frac{\mu_f}{2} \|e_{x,i}^{k+1}\|^2. \tag{39}$$

Combining with Equation (37) gives $\|e_{x,i}^{k+1}\|^2 \leq \frac{2}{\mu_f}\epsilon_k^{(i)}$. Likewise, strong convexity in $z$ gives $\|e_z^{k+1}\|^2 \leq \frac{2}{\mu_\phi}\eta_k$ for some $\mu_\phi > 0$.

We derive a blockwise descent inequality for the inexact updates $(x_i^{k+1}, z^{k+1})$, by comparing them with the exact block minimizers $\widehat{x}_i^{k+1} \in \arg\min_{x_i} L(x_{1:i-1}^{k+1}, x_i, x_{i+1:n}^k, z^k, w^k)$ and $\widehat{z}^{k+1} \in \arg\min_z L(x^{k+1}, z, w^k)$.

Fix $i \in \{1, \ldots, n\}$ and define the univariate subproblem

$$\psi_i(u) := L(x_{1:i-1}^{k+1}, u, x_{i+1:n}^k, z^k, w^k).$$

Under Equation (38), we also have

$$\psi_i(x_i^{k+1}) \leq \min_u \psi_i(u) + \epsilon_k^{(i)} = \psi_i(\widehat{x}_i^{k+1}) + \epsilon_k^{(i)}. \tag{40}$$

Combining Equation (39) and Equation (40) gives

$$\|x_i^{k+1} - \widehat{x}_i^{k+1}\|^2 \leq \frac{2}{\mu_f}\epsilon_k^{(i)}. \tag{41}$$

Now compare the *values* across one $x$-block update. Then,

$$L(x_{1:i}^{k+1}, x_{i+1:n}^k, z^k, w^k) - L(x_{1:i-1}^{k+1}, x_i^k, x_{i+1:n}^k, z^k, w^k)$$
$$= \psi_i(x_i^{k+1}) - \psi_i(x_i^k)$$
$$\leq \left(\psi_i(\widehat{x}_i^{k+1}) - \psi_i(x_i^k)\right) + \left(\psi_i(x_i^{k+1}) - \psi_i(\widehat{x}_i^{k+1})\right) \tag{42}$$
$$\leq -\frac{\mu_f}{2}\|\widehat{x}_i^{k+1} - x_i^k\|^2 + \epsilon_k^{(i)}, \tag{43}$$

where the last line uses the exact-case one-block descent.

To express the decrease in terms of the *inexact* step size $\|x_i^{k+1} - x_i^k\|$, we have

$$\|\widehat{x}_i^{k+1} - x_i^k\|^2 \geq \frac{1}{2}\|x_i^{k+1} - x_i^k\|^2 - \|x_i^{k+1} - \widehat{x}_i^{k+1}\|^2 \geq \frac{1}{2}\|x_i^{k+1} - x_i^k\|^2 - \frac{2}{\mu_f}\epsilon_k^{(i)}.$$

Plugging this into Equation (43) we obtain

$$L(x_{1:i}^{k+1}, x_{i+1:n}^k, z^k, w^k) - L(x_{1:i-1}^{k+1}, x_i^k, x_{i+1:n}^k, z^k, w^k) \leq -\frac{\mu_f}{4}\|x_i^{k+1} - x_i^k\|^2 + 2\epsilon_k^{(i)}. \tag{44}$$

Following the same process for block $z$ yields

$$L(x^{k+1}, z^{k+1}, w^k) - L(x^{k+1}, z^k, w^k) \leq -\frac{\mu_\phi}{2}\|\widehat{z}^{k+1} - z^k\|^2 + \eta_k,$$

and

$$\|z^{k+1} - \widehat{z}^{k+1}\|^2 \leq \frac{2}{\mu_\phi}\eta_k.$$

for some $\beta > 0$ from the exact-case $z$-block descent. As in the $x$-block,

$$\|\widehat{z}^{k+1} - z^k\|^2 \geq \frac{1}{2}\|z^{k+1} - z^k\|^2 - \|z^{k+1} - \widehat{z}^{k+1}\|^2 \geq \frac{1}{2}\|z^{k+1} - z^k\|^2 - \frac{2}{\mu_\phi}\eta_k,$$

hence

$$L(x^{k+1}, z^{k+1}, w^k) - L(x^{k+1}, z^k, w^k) \leq -\frac{\mu_\phi}{4}\|z^{k+1} - z^k\|^2 + 2\eta_k. \tag{45}$$

The dual update is $w^{k+1} = w^k + \rho\left(A(x^{k+1}) + Qz^{k+1}\right)$. By linearity of $L$ in $w$,

$$L(x^{k+1}, z^{k+1}, w^{k+1}) - L(x^{k+1}, z^{k+1}, w^k) = \langle w^{k+1} - w^k, A(x^{k+1}) + Qz^{k+1}\rangle$$
$$= \rho\|A(x^{k+1}) + Qz^{k+1}\|^2 = \frac{1}{\rho}\|w^{k+1} - w^k\|^2. \tag{46}$$

Furthermore, we have

$$\|w^{k+1} - w^k\|^2 \leq \frac{\|Q^T w^{k+1} - Q^T w^k\|}{\lambda_{min}^+(Q^T Q)},$$

$$\leq \frac{\|\nabla_z \phi(z^{k+1}) - \nabla_z L(x^{k+1}, z^{k+1}, w^k) - \nabla_z \phi(z^k) + \nabla_z L(x^k, z^k, w^{k-1})\|^2}{\lambda_{min}^+(Q^T Q)},$$

$$\leq \frac{3\|\nabla_z \phi(z^{k+1}) - \nabla_z \phi(z^k)\|^2 + 3\|\nabla_z L(x^{k+1}, z^{k+1}, w^k)\|^2 + 3\|\nabla_z L(x^k, z^k, w^{k-1})\|^2}{\lambda_{min}^+(Q^T Q)},$$

$$\leq \frac{3L_\phi^2}{\lambda_{min}^+(Q^T Q)}\|z^{k+1} - z^k\|^2 + \frac{6(L_\phi + \lambda_{max}(Q^T Q))}{\lambda_{min}^+(Q^T Q)}(\eta_k + \eta_{k-1}) \tag{47}$$

As a result, we get

$$L(x^k, z^k, w^k) - L(x^{k+1}, z^{k+1}, w^{k+1})$$

$$\geq \frac{\mu_f}{4}\|x^{k+1} - x^k\|^2 - 2\sum_i \epsilon_k^{(i)} + \frac{\mu_\phi}{4}\|z^{k+1} - z^k\|^2 - 2\eta_k - \frac{1}{\rho}\|w^{k+1} - w^k\|^2$$

$$\geq \frac{\mu_f}{4}\|x^{k+1} - x^k\|^2 - 2\sum_i \epsilon_k^{(i)} + \left(\frac{\mu_\phi}{4} - \frac{3L_\phi^2}{\rho\lambda_{min}^+(Q^T Q)}\right)\|z^{k+1} - z^k\|^2 - 2\eta_k$$

$$- \frac{6(L_\phi + \lambda_{max}(Q^T Q))}{\rho\lambda_{min}^+(Q^T Q)}(\eta_k + \eta_{k-1})$$

$$\geq \frac{\mu_f}{4}\|x^{k+1} - x^k\|^2 - 2\sum_i \epsilon_k^{(i)} + \frac{\mu_\phi}{8}\|z^{k+1} - z^k\|^2 - 2\eta_k - \frac{6(L_\phi + \lambda_{max}(Q^T Q))}{\rho\lambda_{min}^+(Q^T Q)}(\eta_k + \eta_{k-1})$$

$$\geq \frac{\mu_f}{4}\|x^{k+1} - x^k\|^2 + \frac{\mu_\phi}{16}\|z^{k+1} - z^k\|^2 + \frac{\mu_\phi \lambda_{min}^+(Q^T Q)}{48L_\phi^2}\|w^{k+1} - w^k\|^2 - M\left(\sum_i \epsilon_k^{(i)} + \eta_k + \eta_{k-1}\right). \tag{48}$$

where we apply Equation (47) in the third and the last inequalities, $M = 2 + \frac{6(L_\phi + \lambda_{max}(Q^T Q))}{\rho\lambda_{min}^+(Q^T Q)}$ is a constant and $\rho$ is large enough.

Summing Equation (48) from $k = 0$ to $K$ and telescoping yields

$$L(x^0, z^0, w^0) - L(x^{K+1}, z^{K+1}, w^{K+1}) + 2M \sum_{k=0}^K \left(\sum_i \epsilon_k^{(i)} + \eta_k\right)$$

$$\geq \sum_{k=0}^K \left[\frac{\mu_f}{4}\|x^{k+1} - x^k\|^2 + \frac{\mu_\phi}{16}\|z^{k+1} - z^k\|^2 + \frac{\mu_\phi \lambda_{min}^+(Q^T Q)}{48L_\phi^2}\|w^{k+1} - w^k\|^2\right]$$

By Assumption 2.3 and the quadratic penalty, $L(x^{K+1}, z^{K+1}, w^{K+1})$ is bounded below. While $\sum_{k=0}^K (\sum_i \epsilon_k^{(i)} + \eta_k) < \infty$ by hypothesis. Hence the nonnegative series

$$\sum_{k=0}^\infty \left[\sum_{i=1}^n \|x_i^{k+1} - x_i^k\|^2 + \|z^{k+1} - z^k\|^2 + \|w^{k+1} - w^k\|^2\right] < \infty,$$

which implies $\|x^{k+1} - x^k\| \to 0$, $\|z^{k+1} - z^k\| \to 0$, and $\|w^{k+1} - w^k\| \to 0$. The blockwise optimality follows as in Theorem 3.1.

If further $\sum_i \epsilon_k^{(i)} + \eta_k \leq C[\|x^{k+1} - x^k\|^2 + \|z^{k+1} - z^k\|^2 + \|w^{k+1} - w^k\|^2]$, where

$$C < \frac{\min\{\frac{\mu_f}{4}, \frac{\mu_\phi}{16}, \frac{\mu_\phi \lambda_{min}^+(Q^T Q)}{48L_\phi^2}\}}{2M},$$

then

$$L(x^0, z^0, w^0) - L(x^{K+1}, z^{K+1}, w^{K+1})$$

$$\geq \sum_{k=0}^{K} [\frac{\mu_f}{4} \|x^{k+1} - x^k\|^2 + \frac{\mu_\phi}{16} \|z^{k+1} - z^k\|^2 + \frac{\mu_\phi \lambda_{min}^+(Q^T Q)}{48 L_\phi^2} \|w^{k+1} - w^k\|^2 - 2M(\sum_i \epsilon_k^{(i)} + \eta_k)],$$

$$\geq \sum_{k=0}^{K} C'(\|x^{k+1} - x^k\|^2 + \|z^{k+1} - z^k\|^2 + \|w^{k+1} - w^k\|^2)$$

$$(49)$$

where $C' > 0$. The rest follows the proof of Theorem 3.1, Theorem 3.2 and Theorem 3.3. □

## E   PROOFS OF SECTION 4

**Corollary E.1.** *Under the assumptions of Corollary 4.1, if the iterates of the ADMM applied to the problem in 6 converge to $(x^\star, z^\star, w^\star)$ with $w^\star = 0$, then the Lagrangian converge linearly, i.e., there exists $c_4 > 1$ such that*

$$L(x^k, z^k, w^k) - L(x^\star, z^\star, w^\star) \leq \mathcal{O}(c_4^{-k}),$$

**Corollary E.2.** *Under the assumptions of Corollary 4.2, if $I_i$s are the indicator functions of some polyhedrals, then,*

$$L(x^k, z^k, w^k) - \min_{(x,z) \in \mathcal{B}(x^k, z^k; r)} L(x, z, w^k) \in \mathcal{O}(c_5^{-k}),$$

*where $c_5 > 1$ and $r > 0$. Furthermore, $(x^\star, z^\star)$ is the local minimum of problem Equation (6).*

PROOFS OF COROLLARY 4.1, COROLLARY E.1, COROLLARY 4.2 AND COROLLARY E.2

In this setting, the corresponding objective functions will be

$$F(x) := f(x) + \sum_{i=0}^{T} I_i(x_i), \quad \phi(z) := \sum_{i=1}^{T} \phi(k'),$$

$$f(x) := f(\mathbf{f}), \quad I_i(x_i) := \sum_{j=1}^{N} I_W(\mathbf{f}_i^j).$$

The corresponding multi-affine quadratic constraints of the locomotion problem is $A(x) + Qz = 0$ in which $Q$ is the identity matrix and

$$A(x) := \begin{bmatrix} \mathbf{k}_{init} \\ A^1(\mathbf{f}) \\ \vdots \\ A^T(\mathbf{f}) \end{bmatrix},$$

in which

$$A^i(\mathbf{f}) := \sum_{j=1}^{N} \left( \left( \mathbf{r}_i^j - \mathbf{c}_{\text{init}} - \dot{\mathbf{c}}_{\text{init}} \, i(\Delta t) - \sum_{i'=0}^{i-2} (i - 1 - i')(\sum_{l=1}^{N} \frac{\mathbf{f}_{i'}^l}{m} + \mathbf{g})(\Delta t)^2 \right) \times \mathbf{f}_i^j \right) \Delta t, \text{ for } i \geq 2.$$

From the robotic problem in Equation (6), we have $C_i = 0$ in $A_2^1$ and $A_2^2$. For $A_2^i$, $i \geq 3$, $C_i$ is actually a sparse matrix. If we consider $r_j^t$ as a constant, then

$$
\begin{aligned}
\mathbf{k}'_{i+1} = \mathbf{k}_{i+1} - \mathbf{k}_i &= \sum_{j=1}^{N} \left( \left( \mathbf{r}_i^j - \mathbf{c}_{\text{init}} - \dot{\mathbf{c}}_{\text{init}} i(\Delta t) - \sum_{i'=0}^{i-2}(i-1-i')(\sum_{l=1}^{N} \frac{\mathbf{f}_{i'}^l}{m} + \mathbf{g})(\Delta t)^2 \right) \times \mathbf{f}_i^j \right) \Delta t \\
&= \sum_{j=1}^{N} \left( \left( \mathbf{r}_i^j - \mathbf{c}_{\text{init}} - \dot{\mathbf{c}}_{\text{init}} i(\Delta t) - \sum_{i'=0}^{i-2}(i-1-i')\mathbf{g}(\Delta t)^2 \right) \times \mathbf{f}_i^j \right) \Delta t \\
&\quad - \sum_{j=1}^{N} \sum_{i'=0}^{i-2} \sum_{l=1}^{N} (i-1-i') \frac{\mathbf{f}_{i'}^l \times \mathbf{f}_i^j}{m} (\Delta t)^3 \\
&= \text{Affine Terms on } \mathbf{X} - \sum_{j=1}^{N} \sum_{i'=0}^{i-2} \sum_{l=1}^{N} (i-1-i') \frac{\mathbf{f}_{i'}^l \times \mathbf{f}_i^j}{m} (\Delta t)^3 \\
&= \text{Affine Terms on } \mathbf{X} - \sum_{j=1}^{N} \sum_{i'=0}^{i-2} \sum_{l=1}^{N} \frac{(\Delta t)^3}{m} (i-1-i') \begin{bmatrix} f_{i',y}^l f_{i,z}^j - f_{i',z}^l f_{i,y}^j \\ -f_{i',x}^l f_{i,z}^j + f_{i',z}^l f_{i,x}^j \\ f_{i',x}^l f_{i,y}^j - f_{i',y}^l f_{i,x}^j \end{bmatrix} \\
&= \text{Affine Terms on } \mathbf{X} - \sum_{j=1}^{N} \sum_{l=1}^{N} \frac{(\Delta t)^3}{m} \begin{bmatrix} \frac{1}{2}\mathbf{f}^T \sum_{i'=0}^{t-2} C_{i',y,i,z}^{j,l} \mathbf{f} \\ \frac{1}{2}\mathbf{f}^T \sum_{i'=0}^{t-2} C_{i',x,i,z}^{j,l} \mathbf{f} \\ \frac{1}{2}\mathbf{f}^T \sum_{i'=0}^{t-2} C_{i',x,i,y}^{j,l} \mathbf{f}. \end{bmatrix}
\end{aligned}
$$
(50)

where $C_{i',y,i,z}^{j,l}$, $C_{i',x,i,z}^{j,l}$ and $C_{i',x,i,y}^{j,l}$ only have 4 non-zero element with its number equals to $i - 1 - i'$ or $-(i - 1 - i')$ at $(i'_{x_1}, i_{x_2})$, $(i_{x_2}, i'_{x_1})$, $(i'_{x_2}, i_{x_1})$ and $(i_{x_1}, i'_{x_2})$, where $(x_1, x_2) \in \{(x, y), (x, z), (y, z)\}$. As a result

$$
\| \sum_{i'=0}^{i-2} C_{i',x_1,i,x_2}^{j,l} \| \leq \| \sum_{i'=0}^{i-2} C_{i',x_1,i,x_2}^{j,l} \|_F = \sqrt{\sum_{i'=0}^{i-2} 4(i-1-i')^2} \leq 2i^{3/2}.
$$

In addition,

$$
\begin{aligned}
\|C_i\| = \| \sum_{j=1}^{N} \sum_{i'=0}^{i-2} \sum_{l=1}^{N} \frac{(\Delta t)^3}{2m} C_{i',x_1,i,x_2}^{j,l} \| &\leq \sum_{j=1}^{N} \sum_{l=1}^{N} \frac{(\Delta t)^3}{2m} \| \sum_{i'=0}^{i-2} C_{i',x_1,i,x_2}^{j,l} \| \\
&\leq \sum_{j=1}^{N} \sum_{l=1}^{N} \frac{2i^{3/2}(\Delta t)^3}{2m} \leq \sum_{j=1}^{N} \sum_{l=1}^{N} \frac{2T^{3/2}(\Delta t)^3}{2m} \\
&\leq \frac{N^2 2T^{3/2}(\Delta t)^3}{2m} \in \mathcal{O}\left( \frac{N^2 T^{3/2}(\Delta t)^3}{m} \right).
\end{aligned}
$$

and $C_i \in \mathbb{R}^{n \times n}$, where $n_x = Nn_z = Nn_c = 3N(T+1)$.

If we can seeking a solution on the time interval $[0, T_{total}]$ and we split it into $T$ discretization, then $T = \frac{T_{total}}{\Delta t}$. In consequence,

$$
n_x = Nn_z = Nn_c = 3N\left( \frac{T_{total}}{\Delta t} + 1 \right) = \Theta\left( N\frac{T_{total}}{\Delta t} \right).
$$
(51)

and

$$
\|C\| = \max_i \|C_i\| \in \mathcal{O}\left( \frac{N^2 (T_{total})^{3/2}(\Delta t)^{3/2}}{m} \right).
$$
(52)

For $d_i$, by denoting $a_i^j = \mathbf{r}_i^j - \mathbf{c}_{\text{init}} - \dot{\mathbf{c}}_{\text{init}} i(\Delta t) - \sum_{i'=0}^{i-2}(i-1-i')\mathbf{g}(\Delta t)^2$, notice that,

$$
\begin{aligned}
\sum_{j=1}^{N} \left( \left( \mathbf{r}_i^j - \mathbf{c}_{\text{init}} - \dot{\mathbf{c}}_{\text{init}} i(\Delta t) - \sum_{i'=0}^{i-2}(i-1-i')\mathbf{g}(\Delta t)^2 \right) \times \mathbf{f}_i^j \right) \Delta t &= \sum_{j=1}^{N} \left( a_i^j \times \mathbf{f}_i^j \right) \Delta t \\
&= \sum_{j=1}^{N} \begin{bmatrix} a_{i,y}^j f_{i,z}^j - a_{i,z}^j f_{i,y}^j \\ -a_{i,x}^j f_{i,z}^j + a_{i,z}^j f_{i,x}^j \\ a_{i,x}^j f_{i,y}^j - a_{i,y}^j f_{i,x}^j \end{bmatrix} \Delta t
\end{aligned}
$$

and

$$|a_{i,z}^j| = |\mathbf{r}_{i,z}^j - \mathbf{c}_{\text{init},z} - \dot{\mathbf{c}}_{\text{init},z}t(\Delta t) - \sum_{i'=0}^{i-2}(i-1-i')\mathbf{g}_z(\Delta t)^2|$$

$$\leq |\mathbf{r}_{i,z}^j| + |\mathbf{c}_{\text{init},z}| + |\dot{\mathbf{c}}_{\text{init},z}i(\Delta t)| + |\sum_{i'=0}^{i-2}(i-1-i')\mathbf{g}_z(\Delta t)^2|$$

$$\leq |\mathbf{r}_{i,z}^j| + |\mathbf{c}_{\text{init},z}| + |\dot{\mathbf{c}}_{\text{init},z}T(\Delta t)| + |\sum_{i'=0}^{T-2}(i-1-i')\mathbf{g}_z(\Delta t)^2|$$

$$\leq |\mathbf{r}_{i,z}^j| + |\mathbf{c}_{\text{init},z}| + |\dot{\mathbf{c}}_{\text{init},z}T_{total}| + |\frac{1}{2}\mathbf{g}_z T_{total}^2| \in \mathcal{O}(T_{total}^2).$$

The bound is same for $|a_{t,x}^j|$ and $|a_{t,y}^j|$. As a result, by choosing $x_1, x_2 \in \{x, y, z\}$,

$$\|d_i\| \leq \sqrt{|a_{i,x_1}^j|^2 + |a_{i,x_2}^j|^2} N\Delta t \in \mathcal{O}(NT_{total}^2\Delta t). \tag{53}$$

From Equation (51), Equation (52), Equation (53), Equation (7), $\|e\| = 0$ and $Q = I$, Equation (34) suffices to require

$$\frac{N^2(T_{total})^{3/2}(\Delta t)^{3/2}}{m} \in \mathcal{O}\left(\min\left\{\frac{\mu_f\sqrt{\mu_\phi}}{\sqrt{\frac{T_{total}}{\Delta t}}L_\phi\left(\sqrt{L_f}\sqrt{N\frac{T_{total}}{\Delta t}} + \sqrt{L_\phi}\sqrt{\frac{T_{total}}{\Delta t}}\right)},\right.\right.$$

$$\frac{\mu_f\sqrt{\mu_\phi}}{\sqrt{L_\phi^3\frac{T_{total}}{\Delta t}}\sqrt{N\frac{T_{total}}{\Delta t}}NT_{total}^2\Delta t},$$

$$\left.\left.\left(\frac{\mu_f\sqrt{\mu_\phi}}{L_\phi^{3/2}N\frac{T_{total}}{\Delta t}N\frac{T_{total}}{\Delta t}}\right)^{1/2}\right\}\right),$$

which is equivalent to

$$\Delta t \in \mathcal{O}\left\{\frac{\mu_f^2\mu_\phi m^2}{L_\phi^2(L_f + L_\phi)N^6T_{total}^5}, \frac{\mu_f\sqrt{\mu_\phi}m}{L_\phi^{3/2}N^{7/2}T_{total}^5}, \frac{\mu_f\sqrt{\mu_\phi}m^2}{L_\phi^{3/2}N^6T_{total}^5}\right\}. \tag{54}$$

Once the Equation (54) holds, the Equation (4) is satisfied, and the conclusion from the proof of Corollary D.2, Theorem 3.2, Theorem 3.3 follows. $\qquad\square$

## F  ADDITIONAL EXPERIMENTS INFORMATION

The simulations are done on a normal laptop with Intel(R) Core(TM) i5-1235U with 16GB of memory.

In the 2D locomotion problem, the horizontal location of the end-effector $r_x$ is switched to $r_x + D$ after $M\Delta t$ time steps, where $D$ and $M$ can be chosen randomly for each step to increase the variability in the motion. The frictions are constrained so that the horizontal location of CoM $c_x$ satisfies $-0.15\text{m} \leq c_x \leq 0.15\text{m}$, and vertical location of CoM $c_z$ satisfies $0.15\text{m} \leq c_z \leq 0.25\text{m}$ distance from the stance foot to the CoM. All the other details can be found in the code in the supplementary material.

We further provide the computation time for the locomotion problem under different $\Delta t$. As $\Delta t$ becomes smaller, the dimension of the problem become larger, which requires more time for the computation.

Table 2: Computation time for different $\Delta t$

| $\Delta t$ | 0.05s | 0.02s | 0.01s | 0.005s | 0.002s | 0.001s |
|---|---|---|---|---|---|---|
| Computation time | 0.60s | 1.51s | 3.29s | 7.31s | 18.54s | 38.82s |

Additionally, Fig. 7 confirms that the friction cone constraints are satisfied throughout the optimization process.

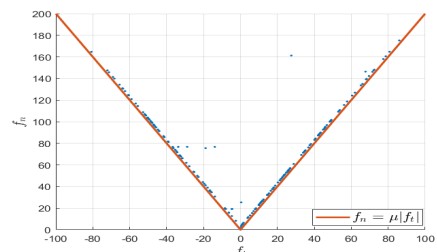

Figure 7: The result of friction and its friction cone.

# G  ADDITIONAL EXPERIMENTS

**Sensitivity of $\rho$:**    Here, we provide additional experiments for the sensitivity analysis of the penalty parameter $\rho$. As illustrated in the left Figure of 8, by increasing the penalty coefficient, the convergence rate decreases while the convergence rate remains linear.

**Approximated ADMM:**    In this experiment, we ran the inexact ADMM, where the updates (i.e., the solutions to the subproblems) are computed using gradient descent (GD) with different numbers of iterations. For example, when the inner-loop iteration count is set to 10, each subproblem is solved using 10 steps of GD. The results are shown in the right panel of Figure 8, where they are compared with the exact ADMM, in which the subproblems are solved analytically.

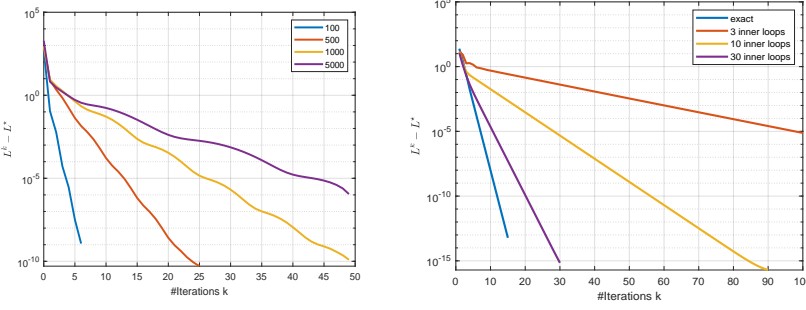

Figure 8: Left: sensitivity analysis of $\rho$. Right: Inexact ADMM with different numbers of GD in the inner loop

**Comparison with other methods:**    Here, we compare several benchmark methods with our algorithm on problems featuring different types of constraints: linear and multi-affine quadratic constraints. These figures contain additional method called IADMM from Tang & Toh (2024). As shown in Figure 9, our ADMM algorithm achieves linear convergence in all settings.

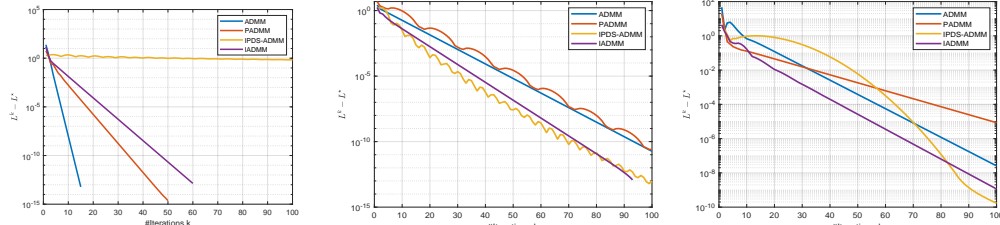

Figure 9: Left: convex objective with multi-affine constraint. Center: convex objective with linear constraint. Right: nonconvex objective with linear constraint

## H    LIMITATION

Our theoretical analysis only establishes convergence of the ADMM for the quadratic constraint problem. The convergence analysis for the higher-order constraints needs further investigation. Also, Theorem 3.2 is satisfied with second order differentiability of the Lagrangian at $(x^\star, z^\star, w^\star)$. A possible relaxed way is to analyze the differential part $x_d^\star$ of $x^\star$, and prove the PL property of $\tilde{L}(x_d, z, w) = L(x_d, x_{-d}^\star, z, w)$, when $x_{-d} = x_{-d}^\star$ is fixed.

