# OpenReview forum: "Last-iterate Convergence of ADMM on Multi-affine Quadratic Equality Constrained Problem"
_ICLR.cc/2026/Conference — Submitted to ICLR 2026_

### Official Review · Reviewer_cBgD · 2025-10-27

**Soundness:** 3
**Presentation:** 2
**Contribution:** 2
**Rating:** 4
**Confidence:** 3

**Summary:**

This paper studies the convergence of ADMM in multi-affine quadratic equality-constrained problems. Under certain assumptions, the main results include:
* A sublinear convergence rate of ADMM with general multi-affine quadratic constraints
* A linear convergence rate of ADMM when the constraints are close to linear constraints

Both results are stated in terms of last-iterate convergence.

Moreover, future results include experiments to explore the effect of multi-affine quadratic constraints on the convergence rate, comparisons with other optimization methods, and applications to locomotion problems in robotics.

**Strengths:**

* The results extend previous linear constraints to affine quadratic constraints, which is more practical than the previous setting where linear constraints are mainly discussed.
* The convergence results are in terms of last-iterate convergence, which is more interesting from a theoretical perspective.
* The discussion of the results in the locomotion problem in robotics is interesting and shows the practical use of the theoretical results.

**Weaknesses:**

My main concern lies in the use of **Assumption 2.3**, which requires that the functions $f(x)$ and $\phi(x)$ are both strongly convex, so the objective function discussed in the current work is a sum of two strongly convex functions and several indicator functions of convex and closed sets. This greatly constrains the degree of non-convexity of the objective function considered in this paper.

Moreover, the authors do not discuss how these convexity assumptions on $f(x)$ compare with those in previous works. For example, in the table at the top of page 3, this seems none of the related works need any convex assumptions on $f(x)$; does this mean that the other works do not require the same convexity assumptions as those used in the current paper?

Other problem:

* In line 97 and the table at the top of page 3, the "KL" should be "PL" as used in Definition 2.5 ?
* In the right part of Figure 5, the labels for each curve are missing.

**Questions:**

* Whether the related works, especially the four related works presented in the table at the top of page 3, also require the strong convexity condition for $f(x)$?
* If they do not, what is the main difficulty of removing this assumption from the current work?
* As the authors stated in the abstract of the current paper, "Although these problems are generally non-convex, they exhibit convexity or related properties when all variables except one are fixed," could the authors provide further discussion on this point? For example, could you provide important real-world examples where such objective functions are used to model problems?

---

> ### Author Response · Authors · 2025-11-19
>
> We sincerely appreciate the reviewer’s constructive comments and thoughtful suggestions. Below, we address the comments and also provide a few additional remarks to further clarify our contributions.
>
> **Regarding the necessity of Assumption 2.3:**
>
>  We would like to emphasize that this assumption enables us to provide stronger types of convergence guarantees for problems with non-linear constraints than those available in prior work, making our results more practical for real-world applications such as the locomotion example.
> Specifically, we would like to **provide last-iterate convergence, explicit convergence rates, and a characterization of the regime**, quantified in terms of the degree of non-convexity $\|C\|$, in which linear convergence can be guaranteed.
> Prior works that impose less restrictive assumptions on the objective, such as smoothness instead of strong convexity, typically require stronger assumptions on the constraints, for example, assuming linear constraints.
> Moreover, they are generally only able to provide guarantees on the average or best iterate instead of last iterate convergence, and these guarantees concern convergence to a stationary point rather than to the optimal point. Please refer to the table in Section 1 for a detailed comparison.
>
>
> Furthermore,  providing such stronger convergence results are crucial for certain applications, particularly in robotics. Specifically, in planning tasks, we often encounter problems with multi-affine constraints. Since planning needs to be both fast and precise, it is essential to use solvers like ADMM that can quickly (linearly) find a solution with predefined accuracy. As a result, factors like convergence rate and the behavior of the last iterate are particularly important.
>
>
> **Regarding the applicability of Assumption 2.3:**
>
> Please note that assuming both functions $f(x)$ and $\phi(z)$ to be strongly convex is not particularly restrictive in real settings such as robotics. For instance, in robotic applications, these functions typically encode energy consumption and/or distance traveled, which are often modeled as quadratic functions and therefore satisfy the strong convexity assumption[1,2].
>
> **Other problems:**
>
> The KL condition is correct and has been used in many literatures. PL condition is a special case of KL condition[3].
> The right Figure in 5 only illustrate the convergence rate for different random initialization, thus, we omitted the labels.
>
> **Question 1:**
>
> This is an important question. While prior work does not require strong convexity of the objective function, it relies on much stronger assumptions on the constraints, for example, linear constraints instead of the non-linear constraints considered in our setting. Moreover, the convergence guarantees provided in these works are weaker and therefore less appealing for real-world applications such as locomotion.
>
> **Question 2:**
>
>  It is important to note that problems with multi-affine quadratic constraints satisfy the property that the constraints become linear when all variables except one are fixed. This property holds in many optimization problems that model physical interactions, such as locomotion or manipulation, once the contact sequences and their timing are specified[4,5].
>
> As discussed in the introduction, solving such problems is crucial for planning. Since planning must be performed online while the robot interacts with its surrounding environment, it is essential to have solvers that provide last-iterate convergence as well as explicit convergence rates, enabling the computation of solutions with a pre-specified precision.
>
> **Reference:**
>
> [1] Villarreal, Octavio, et al. (2020) MPC-based controller with terrain insight for dynamic legged locomotion. 2020 IEEE International Conference on Robotics and Automation (ICRA). IEEE.
>
> [2] Michael Posa, Cecilia Cantu, and Russ Tedrake(2014). A direct method for trajectory optimization of rigid bodies through contact. The International Journal of Robotics Research, 33(1):69–81.
>
> [3] Fatkhullin, I., Etesami, J., He, N., & Kiyavash, N. (2022). Sharp analysis of stochastic optimization under global kurdyka-lojasiewicz inequality. Advances in Neural Information Processing Systems, 35, 15836-15848.
>
> [4] Meduri, A., Shah, P., Viereck, J., Khadiv, M., Havoutis, I., & Righetti, L. (2023). Biconmp: A nonlinear model predictive control framework for whole body motion planning. IEEE Transactions on Robotics, 39(2), 905-922.
>
> [5]Viereck, J., & Righetti, L. (2021). Learning a centroidal motion planner for legged locomotion. In 2021 IEEE International Conference on Robotics and Automation (ICRA) (pp. 4905-4911). IEEE.

---

> > ### Comment · Reviewer_cBgD · 2025-11-26
> >
> > I thank the authors for their explanation.
> >
> > * **Regarding to Assumption 2.3**: In Table 1, I suggest that the authors explicitly state their strong convexity assumption on the objective functions to provide a more comprehensive and fair comparison with related works. In Li et al. (2024), the authors clearly indicate that their results require convexity. However, the current manuscript also relies on a convexity assumption but does not mark this in the comparison table, which makes the comparison somewhat misleading. I recommend revising Table 1 to reflect this requirement.
> >
> > * **KL condition**: I am still confused about this condition. On line 97, the authors state that the definition of the KL condition is provided in Definition 2.5. However, in Definition 2.5, the authors only define the α-PL property. Are these two notions intended to be the same, or is there a distinction that should be clarified?

---

> ### Author Response · Authors · 2025-11-27
>
> We thank the reviewer for the constructive suggestion. We have updated  Table 1 and clearly stated our convexity assumptions.
>
> **KL condition:** This is a keen observation by the reviewer. The KL and $\alpha$-PL conditions are different in general: the KL property is more general, while the $\alpha$-PL condition is a special case of it. To improve clarity, we have added the definition of the KL property in the Appendix (Definition A.4) and emphasized its relationship to the $\alpha$-PL condition.
> Specifically, when $\phi(x)=Cx^{(\alpha-1)/\alpha}$ in the definition of the KL property, where $C$ is a positive constant, then it reduces to the $\alpha$-PL condition given in Definition 2.5.
>
> If there are any other concerns for this paper, please let us know.

---

### Official Review · Reviewer_ViAF · 2025-10-29

**Soundness:** 3
**Presentation:** 3
**Contribution:** 2
**Rating:** 4
**Confidence:** 4

**Summary:**

This paper analyses the convergence properties of the Alternating Direction Method of Multipliers (ADMM) when applied to multi-affine, quadratic, equality-constrained optimisation problems. Under assumptions including $L$-smooth and strongly convex objectives, as well as full-rank constraint matrices, the authors prove that ADMM converges to a stationary point at a rate of at least sublinear. When the nonlinearity in the constraints is sufficiently small relative to the linear components, they also prove linear convergence. These theoretical findings are applied to robotic locomotion trajectory optimisation, where centroidal dynamics lead to multi-affine constraints, and are experimentally validated.

**Strengths:**

- This paper clearly demonstrates the real-world applicability of centroidal dynamics in locomotion by highlighting how they give rise to multi-affine quadratic constraints.
- It provides comprehensive theoretical results and several meaningful extensions.
- The analysis makes novel contributions by establishing explicit conditions for linear convergence when nonlinear constraint coefficients are sufficiently small in relation to linear components.

**Weaknesses:**

The author needs to make a major revision to improve the quality. Some detailed comments are provided below.

- Although multi-affine quadratic constraints are generally challenging, the problem becomes affine when all but one variable block is fixed. ADMM can naturally exploit this structure, which appears to have a negligible effect on the analysis framework of classic ADMM. The convergence rate (Theorem 3.1) also basically follow the convergence analysis in [1] and the KL framework in [2]. (Here the PL property is a special case of KL).
- Several extensions of the main results appear incomplete:
  - Although the main motivation is robotic applications, Corollaries 4.1 and 4.2 only guarantee convergence when certain assumptions about the functions are met. This means that, although problem (6) can be reformulated as (1), theoretical convergence is not fully assured, thereby limiting its practical application.
  - The analysis of the approximated-ADMM (Algorithm 2) does not explicitly address the effect of approximation errors on convergence. Furthermore, in Theorem D.3, conditions P2 and P3 are assumed rather than proven, which weakens the theoretical support.
- Although each ADMM subproblem is strongly convex and smooth, a sufficiently large penlty parameter $\rho$ causes the term $\frac{\rho}{2}\|A(x)+Qz\|^2$ in the augmented Lagrange function to dominate. This increases the condition number of the Hessian of the subproblem, which can slow convergence or require high-precision solvers.
- The experimental evaluation lacks rigour and comprehensive baseline comparisons. Comparisons with existing methods (PADMM and IPDS-ADMM are designed for non-convex objectives) are limited to a few scenarios and omit standard benchmarks. Furthermore, since Algorithm 1 is only a classical ADMM, it should also be compared with ADMM methods designed for convex objectives [3,4].
- The paper provides incomplete practical guidance on parameter selection. While a 'sufficiently large $\rho$' is theoretically required, the paper provides no sensitivity analysis or practical tuning strategies, which hinders reproducibility.
- Some of the expressions are inaccurate. For example, on page 3, the description of [5] incorrectly states that $\phi$ is not smooth, and the objective in [6] does not match the original reference.
- It seems that the author is missing some key references, including [7,8], which also address nonconvex optimization with nonlinear equality constraints using ADMM. Since the constraints studied here are a special case of nonlinear equalities, it would be good to contextualize the work the authors have done with these important papers.
- The writing suffers from typographical and organizational issues:
  - The cross-ref to expressions are inconsistent (for example, 'Equation (13)' on page 24, line 1244 vs. 'equation 34' on line 1260).
  - The table on page 3 lacks a caption.
  - On page 3, line 155, 'Assumption' should be pluralized as 'Assumptions'.

# References

[1] Gao, W., Goldfarb, D., & Curtis, F. E. (2020). ADMM for multiaffine constrained optimization. Optimization Methods and Software, 35(2), 257-303.\
[2] Guo, K., Han, D. R., & Wu, T. T. (2017). Convergence of alternating direction method for minimizing sum of two nonconvex functions with linear constraints. International Journal of Computer Mathematics, 94(8), 1653-1669.\
[3] Cai, X., Han, D., & Yuan, X. (2017). On the convergence of the direct extension of ADMM for three-block separable convex minimization models with one strongly convex function. Computational Optimization and Applications, 66(1), 39-73.\
[4] Tang, T., & Toh, K. C. (2024). Self-adaptive ADMM for semi-strongly convex problems. Mathematical Programming Computation, 16(1), 113-150.\
[5] Li, J., Ma, S., & Srivastava, T. (2024). A Riemannian alternating direction method of multipliers. Mathematics of Operations Research.\
[6] Yuan, G. (2025). ADMM for nonconvex optimization under minimal continuity assumption. ICLR.\
[7] El Bourkhissi, L., & Necoara, I. (2025). Convergence rates for an inexact linearized ADMM for nonsmooth nonconvex optimization with nonlinear equality constraints. Computational Optimization and Applications, 1-39.\
[8] Li, B., & Yuan, Y. X. (2025). Convergent Proximal Multiblock ADMM for Nonconvex Dynamics-Constrained Optimization. arXiv preprint arXiv:2506.17405.

**Questions:**

Please see **Weakness**.

---

> ### Author Response · Authors · 2025-11-19
>
> We would like to thank the reviewer for the constructive comments and suggestions. In what follows, we provide several additional remarks to further clarify our contributions.
>
> **Challenges of the Multi-Affine Constrained Problem:**
>
> We emphasize that although the problem becomes affine when all but one variable are fixed, the underlying theoretical challenges remain twofold: **establishing the convergence rate and ensuring local optimality**. As discussed in the Introduction section, the convergence rate of ADMM ranges from sublinear to linear depending on the degree of nonlinearity in the constraint set. In this work, our goal is to quantify this relationship and provide a precise characterization of when such behavior occurs. Achieving this required a more careful and refined analysis of ADMM.
>
> The next challenge is to establish optimality guarantees for the convergence point. Unlike related works that provide only average-iterate or best-iterate convergence results when the constrainst are non-linear, our goal is to show that ADMM actually converges in the last iterate to a local optimal point even when the constraint set is nonlinear (multi-affine). Moreover, we prove that the sequence of iterates itself converges to a local optimum, an outcome not achieved in prior work on nonlinear constrained problems [1, 2].
>
> Please note that both of these results are crucial for certain applications, particularly in robotics. Specifically, in planning tasks, we often encounter problems with multi-affine constraints. Since planning needs to be both fast and precise, it is essential to use solvers like ADMM that can quickly (linearly) find a solution with predefined accuracy. As a result, factors like convergence rate and the behavior of the last iterate are particularly important.
>
> **Applicability of Assumption 2.3 about the objective function:**
>
> Please note that assuming both functions $f(x)$ and $\phi(z)$ to be strongly convex is not particularly restrictive in real-world settings. For instance, in robotic applications, these functions typically encode energy consumption and/or distance traveled[3,4], which are often modeled as quadratic functions and therefore satisfy the strong convexity assumption as their Hessian is positive definite.
> Thus, when the problem in (6) is instantiated with the standard quadratic costs used in our experiments and in most robotics applications, all assumptions of Corollaries 4.1 and 4.2 are satisfied, and the convergence guarantees do apply directly to problem (6).
>
> **Regarding the Approximated-ADMM:**
>
> We again thank the reviewer for the opportunity to clarify these points. Assumptions P2 and P3 in Theorem D.3 specify conditions under which the inexact subproblem solver produces sufficiently accurate solutions to ensure the overall convergence guarantees. Whether such a solver exists, and how many iterations it requires to satisfy both P2 and P3, is not the primary focus of our work, just as it is not the main focus of other related studies on ADMM, e.g.,  Wang et al.[5]; Barber \& Sidky
> [6] and El Bourkhissi \& Necoara[1]. Nonetheless, the answer is positive, such solvers do exist; for example, gradient descent can be used to meet these assumptions.
>
> To clarify more, in Section D of the appendix, we present a new theorem, Theorem D.4 (highlighted in blue), which quantifies the influence of the approximation error introduced by an inexact subproblem solver.
> The convergence guarantees follow from a standard perturbation of the exact descent inequality.
> Additionally, we conducted new experiments to validate these results, which are presented in Section G of the Appendix, Figure 8. In this figure, we show the convergence behavior of inexact ADMM when applying gradient descent with a finite number of iterations for the sub-problems. As illustrated, the convergence remains linear, demonstrating the effectiveness of inexact ADMM.
>
> **Regarding the penalty parameter $\rho$:**
>
>  We would like to emphasize that the requirement that $\rho$ should be chosen “sufficiently large” is common in ADMM theory for nonconvex or multi-block settings.
> To be more rigorous, we provide a lower bound on the choice of $\rho$ in Theorem 3.1.
> Additionally, as suggested by the reviewer, we performed further experiments to analyze the sensitivity of $\rho$ and present the results in Section G of the Appendix.
> As shown in Figure 8, the convergence rate remains efficient even for large value of $\rho=5000$.

---

> ### Author Response · Authors · 2025-11-19
>
> **About the related baselines:**
>
> Please noted that the algorithm in [7] is also classic ADMM, which we already compared against in our previous experiments.
> Also note that the algorithm in [8] cannot be directly applied to multi-affine quadratic equality problem, as it is required to find global minimum for the nonconvex function in the subproblem of algorithm.
> Nonetheless, we extended our comparisons to also include [7], focusing on finding the stationary point rather than the global minimum. The results are presented in Section G, Figure 9. We plan to move this figure to the main text in the final version.
>
> **Typos and Missing References:**
>
> We appreciate the reviewer for pointing out the typos, as well as the missing references.
> We have revised and updated the rebuttal version accordingly, with all newly added materials highlighted in blue.
>
> **References:**
>
> [1] El Bourkhissi, L., \& Necoara, I. (2025). Convergence rates for an inexact linearized ADMM for nonsmooth nonconvex optimization with nonlinear equality constraints. Computational Optimization and Applications, 1-39.
>
> [2] Li, B., \& Yuan, Y. X. (2025). Convergent Proximal Multiblock ADMM for Nonconvex Dynamics-Constrained Optimization. arXiv preprint arXiv:2506.17405.
>
> [3] Villarreal, Octavio, et al. (2020) MPC-based controller with terrain insight for dynamic legged locomotion. 2020 IEEE International Conference on Robotics and Automation (ICRA). IEEE.
>
> [4] Michael Posa, Cecilia Cantu, and Russ Tedrake(2014). A direct method for trajectory optimization of rigid
> bodies through contact. The International Journal of Robotics Research, 33(1):69–81.
>
> [5] Yu Wang, Wotao Yin, and Jinshan Zeng (2019). Global convergence of admm in nonconvex nonsmooth optimization. Journal of Scientific Computing, 78:29–63.
>
> [6] Rina Foygel Barber and Emil Y Sidky. Convergence for nonconvex admm, with applications to ct imaging. Journal of Machine Learning Research, 25(38):1–46, 2024.
>
> [7] Cai, X., Han, D., & Yuan, X. (2017). On the convergence of the direct extension of ADMM for three-block separable convex minimization models with one strongly convex function. Computational Optimization and Applications, 66(1), 39-73.
>
> [8] Tang, T., & Toh, K. C. (2024). Self-adaptive ADMM for semi-strongly convex problems. Mathematical Programming Computation, 16(1), 113-150.

---

### Official Review · Reviewer_yW3e · 2025-10-29

**Soundness:** 4
**Presentation:** 4
**Contribution:** 3
**Rating:** 6
**Confidence:** 3

**Summary:**

This paper shows that the classical ADMM procedure for optimization under equality constraints does converge to a local minimum when the constraints are multi-affine quadratic and the objective function satisfies some additional properties (including strong convexity plus some indicator functions). It further establishes a linear convergence rate when some additional assumptions are satisfied and shows applications to robotic locomotion.

**Strengths:**

The results obtained appear to be new: in particular, the convergence of ADMM under certain assumptions was proven in Guo et al. (2020) but without a convergence rate analysis. The simulation experiments with robots are limited but rather convincing.

I am not an optimization specialist, but the paper is interesting and well written, and a cursory look at the proofs indicates that they are reasonable (e.g., they go further than noting that the sequences are decreasing and bounded below, or that the difference between iterates converges to zero, which would not be sufficient).  Given the wide use of ADMM in the community I think the paper is of interest to the ICLR community

**Weaknesses:**

Although the robotic experiments validate the assumptions made in the paper, it would be nice to discuss these  further, for example Assumption 2.3 on the objective function, which seems rather restrictive, as well as their importance in practice.

**Questions:**

See weaknesses above.

---

> ### Author Response · Authors · 2025-11-19
>
> We would like to thank the reviewer for the positive feedback on our work. To clarify the applicability of Assumption 2.3, we note that assuming both functions $f(x)$ and $\phi(z)$ to be strongly convex is not particularly restrictive in real settings such as robotics. For instance, in robotic applications, both functions typically encode energy consumption and/or distance traveled, which are often modeled as quadratic functions and therefore satisfy the strong convexity assumption.
>
> Another important remark regarding the necessity of this assumption concerns the type of convergence guarantees we aim to establish in our work. Specifically, we would like to provide last-iterate convergence, explicit convergence rates, and a characterization of the regime, quantified in terms of the degree of non-convexity $\|C\|$, in which linear convergence can be guaranteed.
> Prior works that impose less restrictive assumptions on the objective, such as smoothness instead of strong convexity, are typically only able to provide guarantees on the average or best iterate, and these guarantees pertain merely to convergence to a stationary point rather than the optimal point. please refer to the table in Section 1.
>
> Please note that providing such stronger convergence results are crucial for certain applications, particularly in robotics. Specifically, in planning tasks, we often encounter problems with multi-affine constraints. Since planning needs to be both fast and precise, it is essential to use solvers like ADMM that can quickly (linearly) find a solution with predefined accuracy. As a result, factors like convergence rate and the behavior of the last iterate are particularly important.

---

> > ### Comment · Reviewer_yW3e · 2025-11-26
> >
> > Thank you for the clarification:

---

### Official Review · Reviewer_w4dv · 2025-10-31

**Soundness:** 3
**Presentation:** 4
**Contribution:** 3
**Rating:** 8
**Confidence:** 3

**Summary:**

The paper proves the convergence of a variant of the alternating direction method of multipliers (ADMM) for multi-affine quadratic equality constrained problems, which are non-convex. Assumptions are less restrictive than in prior work. Linear convergence rates are also proven. The ADMM scheme is evaluated on robotics locomotion problems.

**Strengths:**

The paper tackles an important and non-trivial problem: designing solvers with convergence guarantees for non-convex problems, with many applications. The paper is well-written throughout. Assumptions and results are clearly stated, discussed, and compared to other ones in the literature, which makes the contribution clear. Examples are instructive and show the necessity of assumptions. The ADMM scheme is tested on a non-trivial locomotion problem, and results validate the derived convergence rates.

**Weaknesses:**

The following limitations and suggestions are minor:

1) Application to locomotion: The proposed method can only handle the case with pre-defined contact sequences and timings. This limitation should be stated.

2) Section 5, baselines:
- Adding a sentence describing the baselines and their difference with the proposed ADMM scheme would strengthen the comparison.
- Computation times for solving the locomotion problem are not reported. It is unclear if the proposed method is faster and converges more robustly in practice than other tailored solvers for such problems.

3) Mathematical clarifications:
- On line 223, the dual variable $w$ has the wrong dimensions and should be in $\mathbb{R}^{n_c}$.
- In Definition 2.1, "such" => "such that". Also, "Moreover," should be replaced with an "and" for the definition to make sense. Also, it could be worth noting that quadratics ($C_i=I$) do not satisfy this assumption, so this assumption implies that the diagonal elements $(C_i)_{jj}$ are zero.
- Typo: in (18), instances of $\nabla A$ should be $\nabla_i A$.
- The first step of the proof of Theorem 3.2 states that second order differentiability of the Lagrangian at $(x^\star,z^\star,w^\star)$ implies strict feasibility in a neighborhood ("indicator functions are all zero for the points in that neighborhood"), which is correct. This assumption implies that $(x^\star,z^\star,w^\star)$ is in the strict interior of the set $\cap_i X_i$. This assumption can be strong and it would be worth discussing how to potentially relax it, e.g., with a refined assumption and analysis accounting for active constraints on the boundary of the $X_i$'s.
- On line 1199, $\alpha=1/2$ should be $\alpha=2$.

**Questions:**

Please clarify Definition 2.1 and the assumption of second order differentiability of the Lagrangian in Theorem 3.2 (see my previous comment).

---

> ### Author Response · Authors · 2025-11-19
>
> We thank the reviewer for the positive evaluation of our paper and appreciate the constructive comments and the identification of typographical errors. We have addressed comments and corrected the typos in the rebuttal version.
>
> The reviewer has a valid point about the locomotion application. To clarify this, we have added "and fixing the contact sequence and its timing." at the beginning of the Application section, Section 4.
>
> We also appreciate the reviewer’s constructive suggestions and their careful identification of the minor error. We have addressed them in the new version and also added a sentence to describe the baselines in the experiments section, highlighted in blue.
> Moreover, additional details about the computation time are now reported in the appendix Section F. In addition, we add a possible way of proving the linear convergence when relaxing second order differentiability of the Lagrangian, which we mention it in the appendix Limitation.

---

> > ### Comment · Reviewer_w4dv · 2025-11-25
> >
> > Thank you for your response, which addressed my questions. As a small additional comment on Definition 2.1, please remove the dot at the end of Equation (2.1) to clarify the sentence.

---

> > > ### Author Response · Authors · 2025-11-26
> > >
> > > Thank you for your suggestion. We have updated the paper. If there is any other concern, please let us know.

---

### Meta-Review · Area_Chair_oM6x · 2026-01-06

**Summary:**

Reviews split on whether the contribution is a substantive advance or an incremental step in view of established nonconvex ADMM results. Two reviewers view the framework and rate characterization as useful and reasonably novel, while two reviewers argue the proof strategy closely follows prior KL-style ADMM arguments and that the guarantees rely on restrictive objective assumptions and a "small nonlinearity" regime, limiting generality. The rebuttal improves clarity and adds supporting appendix material, but in my opinion concerns remain regarding novelty.

**Reviewer Concerns:**

Novelty. Analysis largely mirrors classic ADMM convergence frameworks and prior work on multiaffine/nonlinear equality constraints, with limited conceptual separation.

Assumptions. Strong convexity and additional regularity requirements are seen as restrictive. Linear-rate claims depend on nonlinearity being sufficiently small. Inexact/approximated ADMM relies on assumed solver conditions.

Experiments. Limited baseline breadth and benchmarking in the main paper, with incomplete runtime and parameter-selection guidance.

**Reviewer Scores:**

Unchanged. viAF and cBgD may increase but unclear whether can push over line.

---

### Decision · Program_Chairs · 2026-01-26

Reject